# The Lin28b/Wnt5a axis drives pancreas cancer through crosstalk between cancer associated fibroblasts and tumor epithelium

Zhaoqi Shu[1,2,11], Minghe Fan[1,2,11], Bo Tu[3,4,11], Zhiheng Tang[5], Haojie Wang[6], Haimeng Li[1,2], Hengchao Li[7], Meng Yuan[1,2], Jingru Bai[8,9], Sihan Huo[1,2], Lina Wang[1], Wei-Guo Zhu[8,9], Wei Wang[10], Xiaoyun Liu [5], Shaokun Shu[2] & Ying Zhao [1,2] ✉

Bidirectional signal transduction between tumor epithelial cells and tumor microenvironment (TME) is important for tumor development. Here we show that Lin28b/let-7 pathway is indispensable for modulating the expression of Wnt5a in tumor epithelium, which could be secreted and then up-regulates Lin28b in cancer-associated fibroblasts (CAFs). Moreover, we demonstrate that Lin28b in CAFs promoted growth of PDAC by inducing cytokine PCSK9's production. Using an orthotopic mouse model of PDAC, we find that depletion of Lin28b in CAFs reduced tumor weight, highlighting the importance of Lin28b in PDAC stroma. Thus, our study shows that the Lin28b-Wnt5a axis plays a critical role in bidirectional crosstalk between pancreatic tumor epithelium and TME and results in a pro-tumorigenic contexture.

Pancreatic ductal adenocarcinoma (PDAC) is characterized by widespread fibrotic desmoplasia, among which cancer-associated fibroblasts (CAFs) are considered one of the most critical stromal cells that interact with tumor epithelium and promote tumor growth, metastasis, and treatment resistance by secreting excessive amounts of ECM proteins, metabolites, cytokines, chemokines, and growth factors[1–3]. Bidirectional signal transduction between cancer cells and CAFs is important for tumor development. Numerous previous studies have shown the critical roles of CAFs in tumorigenesis and development, while some noteworthy effects of tumor cells on CAFs have also been identified. For example, TGF-β is regarded as a major tumor cell-derived factor affecting CAFs' activation[4] and pro-metastatic factors production[5]. PDAC-derived exosomes modulate the liver

microenvironment and promote metastasis by stimulating insulin-like growth factor 1 (IGF-1) signaling in hepatic satellite cells (HSCs)[6]. Extracellular vesicles (EVs) delivery of miRNA in tumor cells can induce the reprogramming of fibroblasts into CAFs[7]. Despite these studies on tumor microenvironment (TME) regulation, more specific roles and detailed mechanisms of tumor cells affecting CAFs' function remain to be further explored.

Lin28b is a RNA-binding protein that promotes cell transformation and is associated with advanced human malignancies[8]. The well-characterized function of Lin28b is to block let-7 miRNA maturation and modulate mRNA translational efficacy by binding to mRNA targets[9–11]. In pancreatic cancer, Lin28b was shown to be capable of promoting the growth and survival of PDAC cells through the

[1]Beijing Key Laboratory of Protein Posttranslational Modifications and Cell Function, Department of Biochemistry and Molecular Biology, School of Basic Medical Sciences, Peking University Health Science Center, Beijing 100191, China. [2]Peking University International Cancer Institute, Peking University Health Science Center, Beijing 100191, China. [3]Molecular and Cellular Oncology Department, the University of Texas MD Anderson Cancer Center, Houston, TX, USA. [4]Clinical Division, Fred Hutchinson Cancer Research Center, Seattle, WA 98109, USA. [5]Department of Microbiology, School of Basic Medical Sciences, Peking University Health Science Center, Beijing, China. [6]Center for Precision Medicine Multi-Omics Research, School of Basic Medical Sciences, Peking University Health Science Center and Peking University Cancer Hospital and Institute, Beijing, China. [7]Department of Pancreatic Surgery, Huashan Hospital, Institute of Pancreatic Disease, FuDan University, Shanghai 200040, China. [8]Marshall Laboratory of Biomedical Engineering, Shenzhen University School of Medicine, Shenzhen 518055, China. [9]Department of Biochemistry and Molecular Biology, School of Medicine, Shenzhen University, Shenzhen, China. [10]Department of Immunology, School of Basic Medical Sciences, Peking University, NHC Key Laboratory of Medical Immunology, Beijing, China. [11]These authors contributed equally: Zhaoqi Shu, Minghe Fan, Bo Tu. ✉e-mail: zhaoying0812@bjmu.edu.cn

inhibition of let-7 and subsequent activation of let-7 target genes[12]. In addition, Lin28b is highly expressed in circulating tumor cells of PDAC patients and functions as a metastasis driver[13]. Although LIN28 is correlated with PDAC growth and metastasis, neither its regulation nor its functional role in PDAC TME has previously been explored.

Here in our study, we find that Lin28b can be expressed in CAFs. In addition, we explore the role of Lin28b in CAFs by using *FSP-Cre;Lin28b*[fl/fl] mice and identify cytokine PCSK9 as a direct target of Lin28b/let-7. This finding highlights the role of Lin28b in stroma.

## Results

### Lin28b could be highly expressed in CAFs of PDAC

Previous studies have shown that LIN28B drives the growth of PDAC cells and elevated LIN28B was correlated with poor prognosis in patients with PDAC[12]. To further explore the function of LIN28B in PDAC, IHC (Immunohistochemistry) staining was performed on human PDAC tissue microarrays (TMAs). We found that 23 of 80 PDAC samples (28.75%) expressed high levels of LIN28B in tumor epithelium. Notably, LIN28B expression was associated with advanced tumor grade and poor prognosis (Fig. 1a–c). Meanwhile, we observed an interesting phenomenon that most LIN28B[high] samples (22 of 23 samples) also exhibited high LIN28B expression in stroma (Fig. 1d), urging us to further explore the expression profiles of LIN28B in PDAC stroma. Cancer-associated fibroblast (CAFs) characterized by α-SMA expression is considered major stromal components in PDAC[14]. Thus, we conducted immunofluorescence staining for α-SMA on TMAs and observed the co-localization of LIN28B and α-SMA in LIN28B[high] samples (Fig. 1e), indicating that LIN28B expression in CAFs was highly correlated with its expression status in tumors. We next sought to validate our findings in mice models of PDAC. Primary murine PDAC lines 14837T, 14838T, and 15376T were isolated from genetically engineered C57BL/6 mice (*p48-Cre;tetO_LSL-Kras*[G12D]*;ROSA_rtTA;p53*[L/+]). We found that Lin28b was highly expressed in 15376T but not in 14837T or 14838T (Fig. 1f, g). Next, we implanted 14837T, 14838T, or 15376T into the pancreas of mice and found that orthotopic model of 15376T expressed high levels of Lin28b in both tumor and stroma, while 14837T and 14838T tumors demonstrated only background levels of staining for Lin28b (Fig. 1h, j). These results suggested that Lin28b could be expressed in CAFs, although Lin28b is generally considered highly expressed in embryonic tissues or cancer tissues[15].

### Lin28b expression in CAFs is induced by conditioned medium from Lin28b[high] tumor cells

We next isolated mouse CAF cell lines (mCAFs, termed 14837CAFs, 14838CAFs, and 15376CAFs) from orthotopic tumors (Supplementary Fig. 1a, b). However, we found that Lin28b was hardly detected in these cell lines cultured in vitro, not consistently with the immunofluorescence data (Fig. 2a, b). We suspected that the PDAC cells might play a role in sustaining stroma Lin28b expression. Thus, mCAFs were co-cultured with murine PDAC lines to mimic the in vivo environment. Interestingly, Lin28b could be induced in CAFs which were co-cultured with 15376T, but not with 14837T or 14838T, suggesting that Lin28b expression in CAFs is related with its expression status in tumors (Fig. 2a, b). To further confirm our hypothesis, 14837CAFs were directly co-cultured with 15376T or 14837T for 6 days and then sorted by flow cytometry (FACS) (Fig. 2c). We found Lin28b expression was increased when 14837CAFs were co-cultured with 15376T but not with 14837T (Fig. 2d). Furthermore, we treated CAFs with conditioned medium (CM) from 15376T for 6 days and also found that Lin28b expression was gradually upregulated by 15376T-CM (Fig. 2e–j and Supplementary Fig. 1c, d). In addition, the ability of 15376T-CM to increase Lin28b expression in CAFs was abolished when Lin28b was knocked-out in 15376T (Fig. 2k, l and

Supplementary Fig. 1e). We next sought to validate our findings in human PDAC. We measured LIN28B expression in human PDAC cell lines and found that LIN28B was highly expressed in PANC-1 and PANC03.27, is not expressed by ASPC-1, Mia Paca-2, PaTu8988T and PaTu8988S (Fig. 2m, n). In addition, we isolated human PDAC cell lines from 2 patients and found that LIN28B was highly expressed in hPDAC1[#] but not in hPDAC2[#] (Fig. 2m, n). Similar data were seen in human PDAC, as CM from PANC-1, PANC03.27 and hPDAC1[#] induced LIN28B expression in human CAFs (hCAFs) in a LIN28B-dependent manner (Fig. 2p–r and Supplementary Fig. 1f–h), confirming that loss of Lin28b in tumor epithelium leads to the inactivation of Lin28b in CAFs in both human and murine PDAC. To verify this phenomenon in vivo, 15376T or Lin28b-KO 15376T were injected into the pancreas of mice. Immunofluorescence staining showed that silencing Lin28b in tumor reduced stromal Lin28b expression in orthotopic models (Fig. 2s). All together, these data indicated that tumor cells with high levels of Lin28b (Lin28b[high]) could upregulate stroma Lin28b expression.

### Lin28b[high] PDAC cells secret Wnt5a to induce Lin28b expression in CAFs

We then utilized RNA-sequencing (RNA-seq) of 15376T, Lin28b-KO 15376T, and 14837T to further understand the mechanisms by which tumor cells upregulate stroma Lin28b expression. 66 upregulated genes were identified in 15376T compared with Lin28b-KO 15376T and 14837T (Fig. 3a). In these genes, we noticed two soluble Wnts (Wnt ligands) including Wnt5a and Wnt10a were upregulated in 15376T (Fig. 3b, c and Supplementary Fig. 2a, b) as well as PANC-1 (Fig. 3d, e and Supplementary Fig. 2c, d). In addition, Wnt5a and Wnt10a were detectable by enzyme-linked immunosorbent assay (ELISA) in CM from Lin28b[high] tumors but not Lin28b[low] tumors (Fig. 3f, g and Supplementary Fig. 2e, f). We then tried to confirm the role of Wnt5a or Wnt10a in driving Lin28b expression in CAFs. As shown in Fig. 3h–l and Supplementary Fig. 2g–m, knocking-out Wnt5a but not Wnt10a in tumors suppressed the ability of tumor cells to induce Lin28b expression in CAFs, highlighting the role of Wnt5a in Lin28b induction. Moreover, we found Lin28b expression in CAFs was upregulated by recombinant-Wnt5a (Fig. 3m–q), while blockade of Wnt5a signaling with neutralizing antibody blunted the ability of 15376T-CM to up-regulate Lin28b expression in CAFs (Fig. 3r, s). We next examined the role of Wnt5a in vivo by injecting 15376T and Wnt5a-KO 15376T into the pancreas of mice. Immunofluorescence staining showed that knocking-out Wnt5a in tumor reduced stromal Lin28b expression in orthotopic models (Fig. 3t).

### Wnt5a-Fzd4-β-catenin pathway plays a critical role in Lin28b induction

Since we provided evidence that Wnt5a leads to the activation of Lin28b in PDAC stroma, we next investigated the mechanism of how Wnt5a induced Lin28b expression. As we known, Wnt signaling is traditionally categorized into two broad pathways, β-catenin-dependent canonical pathway and β-catenin-independent non-canonical pathway. Wnt5a has traditionally been known as a non-canonical WNT. However, in addition to activation of the non-canonical signaling pathways, Wnt5a was reported to activate the canonical pathway in the presence of Fzd4 receptors[16,17]. Therefore, we tested *Fzd4*'s level and found that *Lin28b* could only be detected in both *Wnt5a* and *Fzd4* high-expressed tumor cells lines (Fig. 4a, b), indicating that intact Wnt5a-Fzd4 pathway is essential for activation of Lin28b in PDAC. Next, we knocked-out Fzd4 in CAFs and found neither 15376T-CM nor recombinant-Wnt5a could induce Lin28b expression in Fzd4-KO CAFs (Fig. 4c, d). To further confirm the importance of Fzd4 and take into account CAF heterogeneity, we isolate CAFs from KPC (*Kras*[LSL-G12D/+]*;Trp53*[R172H/+]*;Pdx1-Cre*) mice using FACS[18] and then used Fzd4 antibody to segregate CAFs into Fzd4-positive and Fzd4-negative populations (Fig. 4e). Similarly,

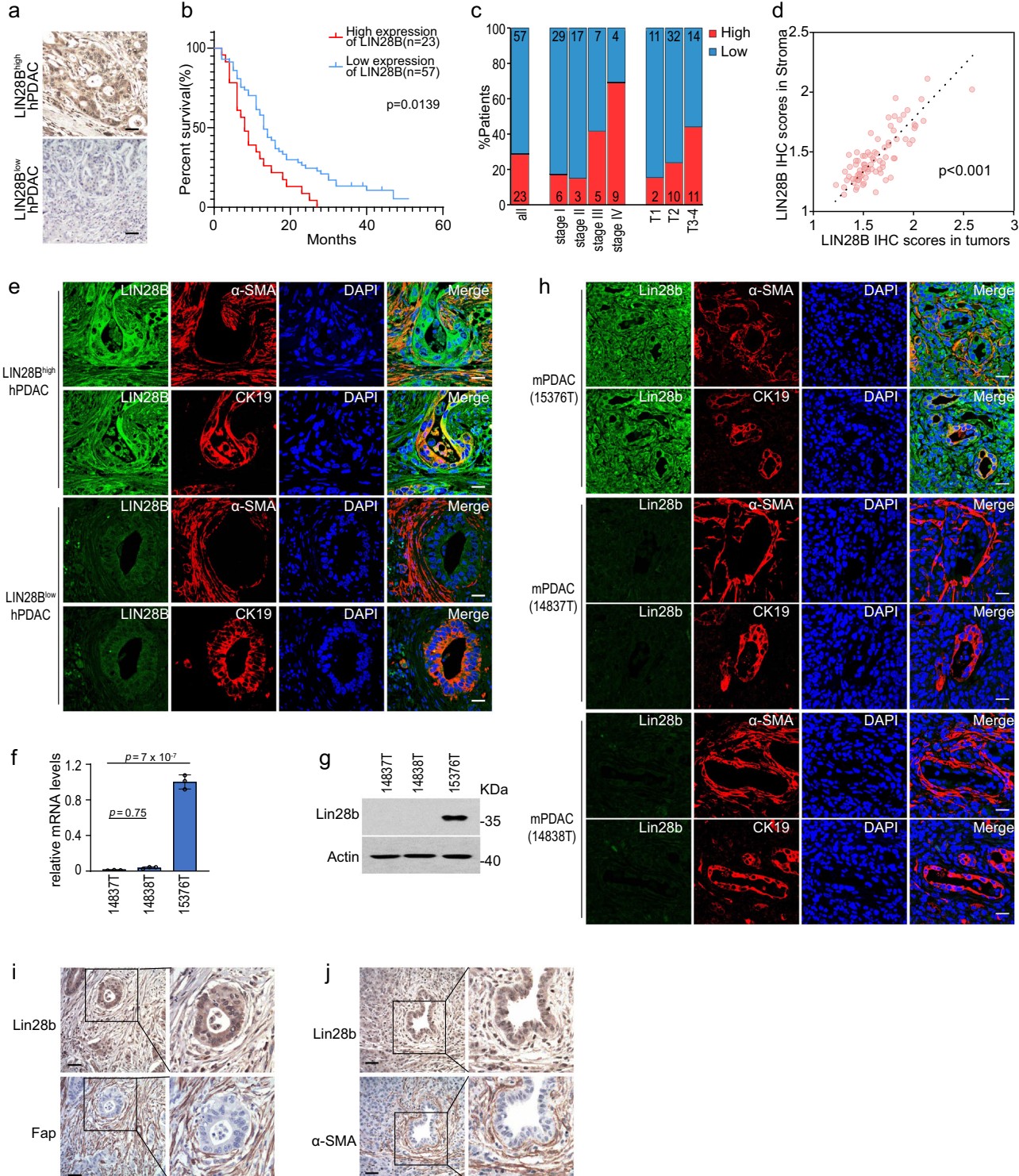

**Fig. 1 | Lin28b could be highly expressed in CAFs of PDAC. a** IHC was performed on a PDAC patient TMA with LIN28B antibody. High expression of LIN28B (LIN28Bhigh) and low expression of LIN28B (LIN28Blow) in tissues was shown. Scale bar: 30 μm. **b** Survival curves for PDAC patients with high (red) or low (blue) levels of LIN28B. Statistical analysis was performed using two-sided Gehan-Breslow-Wilcoxon test; $P = 0.0139$; $n = 80$ patients. **c** Percentages of total population of patients ($n = 80$) or of patients at each pathological stage (stage I; $n = 35$, stage II; $n = 20$, stage III; $n = 12$, stage IV; $n = 13$) and histological T status (T1; $n = 13$, T2; $n = 42$, T3-4; $n = 25$) expressing high (red) or low (blue) levels of LIN28B. **d** Correlation between LIN28B IHC scores in stroma and tumors (Pearson product-moment correlation test; $r = 0.9962$, $p < 0.001$). **e** Immunofluorescence was performed on the PDAC patient TMA using LIN28B, α-SMA, and CK19 antibodies. Representative images from LIN28Bhigh hPDAC and LIN28Blow hPDAC were shown. Scale bar: 30 μm. **f, g** The levels of Lin28b in 14837T, 14838T, and 15376T were measured by real-time qPCR (**f**) and western blotting (**g**). *P*-value by one-way ANOVA with Tukey's multiple comparison test. Representative of $n = 3$ independent experiments (**g**). **h** Orthotopic PDAC tumors generated with 14837T, 14838T, or 15376T were analyzed by Lin28b, α-SMA, and CK19 immunofluorescence staining ($n = 6$ mice). Representative images are shown. Scale bar: 30 μm. **i** Lin28bhigh PDAC tumors were stained for Lin28b, Fap (**i**), and α-SMA (**j**) ($n = 6$ mice). Representative images are shown. Scale bar: 30 μm. Three biologically independent experiments were performed (**f**). Data are shown as mean ± s.d.

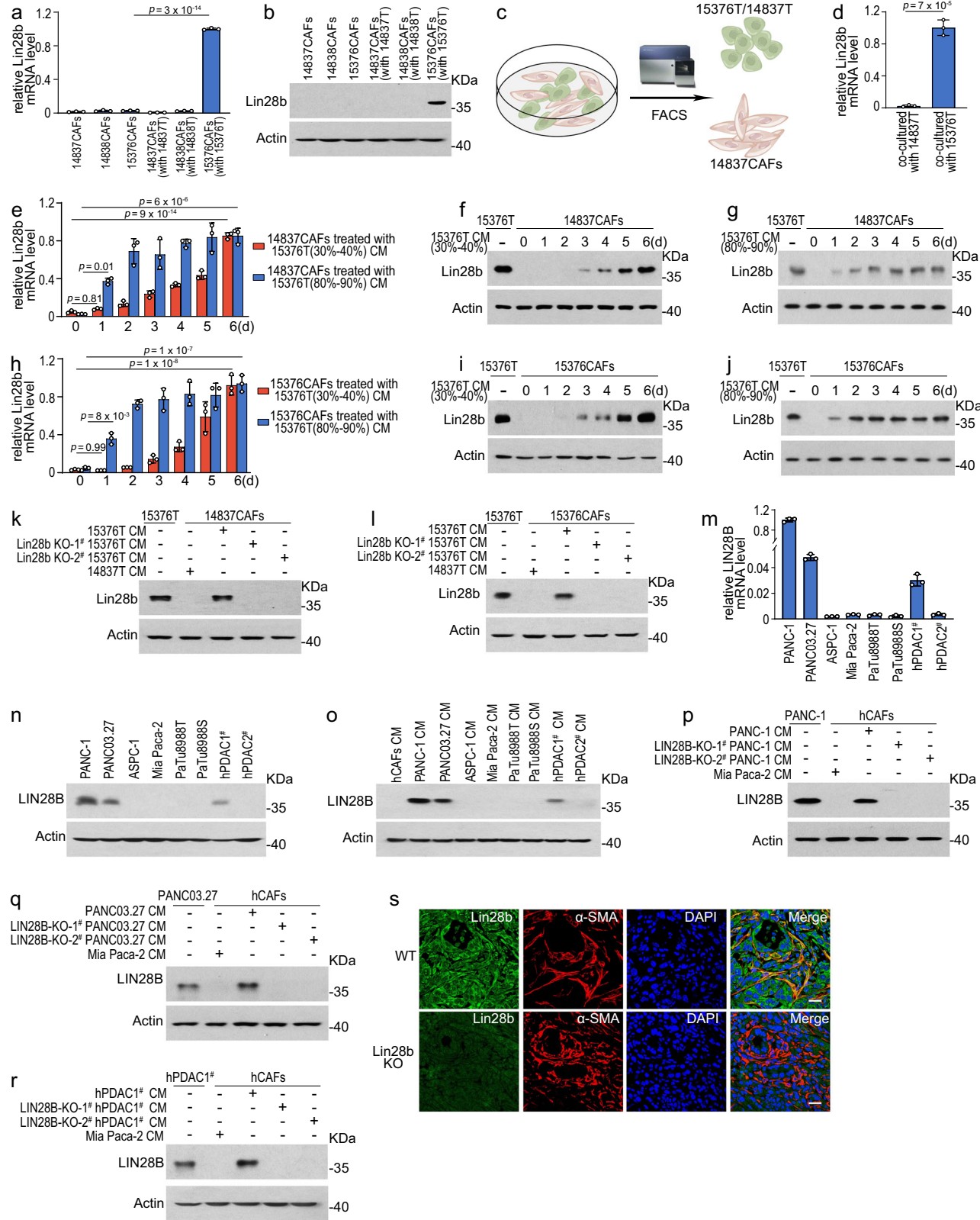

Lin28b expression could be induced only in Fzd4-positive CAFs (Fig. 4f). We also found the levels of β-catenin, an intracellular signal transducer in the Wnt pathway, were upregulated in CAFs when Lin28b expression was induced (Fig. 4g–i). Moreover, both Lin28b^high tumors-CM and r-Wnt5a can induce enrichment of β-catenin in Lin28b promoters (Fig. 4j–l) in CAFs. To further assay Wnt/β-catenin signaling pathway activation, we transfected T-cell factor/lymphoid enhancer-binding factor (TCF/LEF) luciferase reporter into CAFs and found that both Lin28b^high tumors-CM and r-Wnt5a promoted the luciferase activity (Fig. 4m), suggesting that Wnt/β-catenin signaling pathway was activated. In addition, inhibition of β-catenin in CAFs attenuated Lin28b expression induced by CM from Lin28b^high tumors (Fig. 4n–s). These data suggested that Wnt5a-Fzd4-β-catenin pathway plays a critical role in stimulating Lin28b expression in CAFs.

**Fig. 2 | Lin28b expression in CAFs is induced by conditioned medium from Lin28b<sup>high</sup> tumor cells. a, b** Real-time qPCR analysis (**a**) and western blotting analysis (**b**) for Lin28b expression in mCAFs cultured alone or co-cultured with indicated tumor cells. Representative of *n* = 3 independent experiments (**b**). **c, d** 14837CAFs were direct co-cultured with 14837T or 15376T for six days and then sorted by flow cytometric cell sorting (FACS) (**c**). The levels of Lin28b were measured by real-time qPCR (**d**). **e–j** 14837CAFs (**e–g**) or 15376CAFs (**h–j**) were cultured with 15376T-CM for up to 6 days. CM was derived from 15376T with low density (30–40%) or high density (80–90%). Cells were collected for real-time qPCR (**e, h**) and Western blotting (**f, g, i, j**) at indicated time points. Lin28b levels in 15376T were included as a positive control. Representative of *n* = 3 independent experiments (**f, g, i, j**). **k, l** 14837CAFs (**k**) or 15376CAFs (**l**) were cultured with 15376T-CM or Lin28b-KO 15376T-CM for 6 days. Then, Lin28b levels were measured by western blotting. Lin28b levels in 15376T were included as a positive control. Representative of *n* = 3 independent experiments. **m, n** The levels of LIN28B in human PDAC cell lines were measured by real-time qPCR (**m**) and western blotting (**n**). Representative of *n* = 3 independent experiments (**n**). **o** human CAFs (hCAFs) were cultured with CM from human PDAC cell lines for 6 days. Then, LIN28B levels were measured by western blotting. Representative of *n* = 3 independent experiments (**p–r**) hCAFs were cultured with PANC-1-CM (**p**), LIN28B-KO PANC-1-CM (**p**), PANC03.27-CM (**q**), LIN28B-KO PANC03.27-CM (**q**), hPDAC1<sup>#</sup>-CM (**r**), and LIN28B-KO hPDAC1<sup>#</sup>-CM (**r**) for 6 days. Then, the levels of LIN28B were measured by western blotting. LIN28B levels in PANC-1 (**p**), PANC03.27 (**q**), and hPDAC1<sup>#</sup> (**r**) were included as a positive control. Representative of *n* = 3 independent experiments. **s** Orthotopic PDAC tumors generated with 15376T or Lin28b-KO 15376T were analyzed by Lin28b and α-SMA immunofluorescence staining (*n* = 6 mice). Representative images are shown. Scale bar: 30 µm. Three biologically independent experiments were performed (**a, d, e, h, m**). Data are shown as mean ± s.d. *P*-values were determined by one-way ANOVA with Tukey's multiple comparison test (**a, e, h, m**) or two-tailed unpaired Student's t-tests (**d**).

## Lin28b-positive CAFs promote growth of pancreatic cancer

We next sought to explore the role of Lin28b in CAFs. It's well known that rapid glucose consumption by tumors results in a glucose-poor microenvironment[19]. Meanwhile, CAFs can secrete various growth factors and metabolites that sustain the proliferation of cancer cells in a nutrition-poor TME[14]. Not surprisingly, we found CAFs had a positive effect on PDAC proliferation under glucose-limiting conditions (2 mM glucose) which recapitulated the austere tumor microenvironment in vivo[20] (Fig. 5a). In contrast, Lin28b-KO CAFs could not rescue PDAC proliferation under low glucose treatment (Fig. 5b and Supplementary Fig. 3a). We also generated 14837CAF and 15376CAF cell lines stable-expressing WT-Lin28b (Supplementary Fig. 3b–e). We then found that CM from CAFs expressing Lin28b was able to rescue tumor growth under glucose-limiting conditions (Fig. 5c, d). Next, tumor cells were co-injected with different types of CAFs into the pancreas of mice. Co-injection of 15376T with Lin28b-KO 15376CAFs, but not with WT 15376CAFs, slowed the tumor growth in vivo (Fig. 5e–g). Moreover, tumor (14837T) growth was significantly increased when co-injected with WT-Lin28b-expressing CAFs, and this increase was significantly attenuated when PDAC cells were co-injected with 14837CAFs or 15376CAFs (Fig. 5h, i). These findings suggested that stromal Lin28b expression may play a key role in PDAC growth. In addition, since our data indicated that Wnt5a-Fzd4 pathway is essential for activation of Lin28b (Fig. 4c, d, f), we also tested the role of Fzd4 in tumor growth. As shown in Fig. 5j–n and Supplementary Fig. 3f, Fzd4-positive CAFs was able to rescue tumor growth both in vitro and in vivo, indicating that Fzd4-mediated Lin28b expression in CAFs promotes PDAC growth.

To further confirm Lin28b's role in CAFs, we generated a fibroblast-specific conditional Lin28b knock-out model by crossing *Lin28b<sup>fl/fl</sup>* mice with *FSP1-Cre* mice. Then we implanted 15376T into the pancreas of wild-type (WT) mice and *FSP-Cre;Lin28b<sup>fl/fl</sup>* mice and performed the immunofluorescence staining on orthotopic tumors. As shown in Fig. 5o, stromal Lin28b was dramatically downregulated in *FSP-Cre;Lin28b<sup>fl/fl</sup>* mice compared to that in WT mice. In an orthotopic model of PDAC, tumors were significantly smaller when grown in *FSP-Cre;Lin28b<sup>fl/fl</sup>* mice compared to WT mice (Fig. 5p). In addition, Ki67-positive cell populations were substantially reduced in *FSP-Cre;Lin28b<sup>fl/fl</sup>* mice (Fig. 5q, r), demonstrating that Lin28b-positive CAFs promoted PDAC growth in vivo.

## Pcsk9 secreted by CAFs promotes PDAC growth

To further investigate the mechanism by which Lin28b-positive CAFs promotes PDAC growth, we subjected CAFs-CM to three freeze-thaw cycles (−80 °C, 60 °C) or heating (100 °C, 15 min) and found that the ability to increase PDAC proliferation was abolished (Fig. 6a). In addition, CAFs-CM was filtered through 3-kDa cutoff columns and the activity of promoting tumor was retained in the >3-kDa fraction of CAFs-CM (Fig. 6b). Thus, we considered cytokines, but not

metabolites, secreted by CAFs played a role in tumor growth. We next performed quantitative secretomics analysis by using CM from 15376CAFs or Lin28b-KO 15376CAFs to identify cytokines that were secreted by Lin28b-positive CAFs (Fig. 6c and Supplementary Fig. 4a, b). Among the secretion proteins which were downregulated in Lin28b-KO CAFs, five genes (Clu, Cxcl5, FN, PGRN, and Pcsk9) were predicted by TargetScan (https://www.targetscan.org) as the direct targets of Lin28b/let-7 (Fig. 6c). To further verify whether these genes are regulated by Lin28b/let-7 pathway, we generated a mouse Lin28b mutant plasmid which failed to suppress let-7 (Fig. 6d, e). The mutant Lin28b contains five mutations (W34A, F43A, F61A, H159A, and H181A) spread across the cold shock domain (CSD) and CysCysHisCys (CCHC) zinc-finger RNA-binding motifs (W36, F45, F63, H137, and H159 in human)[21]. Next, the CRISPR/Cas9-resistant wild-type Lin28b (flag-Lin28b-WT(r)) and mutant Lin28b (flag-Lin28b-MU(r)) were over-expressed in Lin28b-null cells and we found flag-Lin28b-WT(r), but not flag-Lin28b-MU(r), increased the levels of Clu, Cxcl5, PGRN, and Pcsk9 (Fig. 6f–j). Furthermore, PGRN and Pcsk9 levels in CAFs-CM were much higher than that in tumor-CM, therefore we focused the function of PGRN and Pcsk9 in CAFs (Fig. 6i, j). Treating of PDAC cells with recombinant protein or neutralizing antibody revealed that Pcsk9, but not PGRN, increased PDAC proliferation under glucose-limiting conditions, indicating the role of Pcsk9 in regulating tumor growth (Fig. 6k, l). To further validate the function of Pcsk9, we knocked out Pcsk9 in CAFs (Fig. 6m). Notably, Pcsk9-KO CAFs cannot rescue tumor growth in glucose-limited medium (Fig. 6n). Consistently with the in vitro data, orthotopic tumors were significantly smaller when PDAC cells were co-injected with Pcsk9-KO CAFs compared to that with wild-type CAFs (Fig. 6o–s). We also examined the efficacy of Pcsk9 inhibitor alirocumab in the orthotopic model and found that alirocumab resulted in significant tumor growth inhibition (Fig. 6t). Altogether, these data suggested that Pcsk9 secreted by Lin28b-positive CAFs promotes PDAC growth.

## Pcsk9 is a direct target of Lin28b/let-7

We next sought to determine whether Pcsk9 is the direct target of Lin28b/let-7. Pcsk9 levels in Lin28b-expressing CAFs-CM or Fzd4-positive CAFs-CM were much higher than that in CM of CAFs not expressing Lin28b or Fzd4-negative CAFs (Fig. 7a–h). We found that flag-Lin28b-WT(r), but not flag-Lin28b-MU(r), increased Pcsk9 protein level which was suppressed by Lin28b knocking-out (Fig. 7i). In addition, Pcsk9 was decreased by let-7a agomir and increased by let-7 sponge, indicating that Pcsk9 was able to be regulated by Lin28b/let-7 pathway (Fig. 7j). Then, dual-luciferase reporter assay showed that let-7a agomir suppressed the luciferase driven by Pcsk9 mRNA wild-type 3'-UTR, but did not affect the luciferase driven by mutant 3'-UTR (Fig. 7k, l), suggesting that Pcsk9 was a direct target of let-7a miRNAs. miRNAs regulate gene expression predominantly by promoting mRNA degradation or inhibiting protein translation[9,10]. To further explore

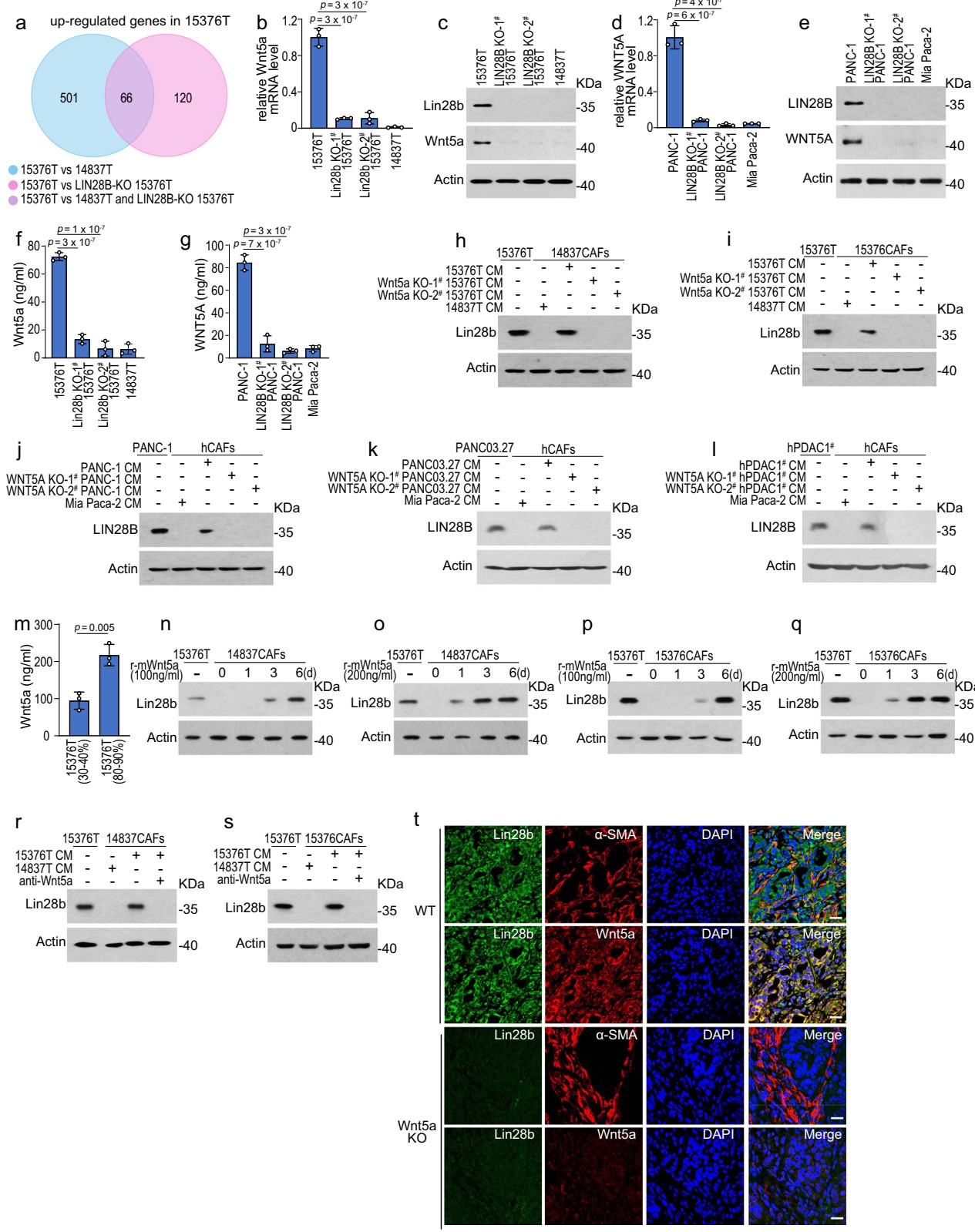

whether Lin28b/let-7 impacts mRNA stability of Pcsk9, we used actinomycin D to block transcription and then detected mRNA levels. *Pcsk9* mRNA decay was not affected by Lin28b knocking-out, suggesting that Lin28b/let-7 didn't regulate *Pcsk9* mRNA degradation (Fig. 7m, n). Thus, we investigated whether Lin28b/let-7 affects Pcsk9 protein level via regulation of translation. Notably, Lin28b depletion reduced the presence of *Pcsk9* mRNA in the polysome fraction but increased in non-translating ribosome fractions (Fig. 7o). Furthermore, flag-Lin28b-WT(r) but not flag-Lin28b-MU(r) induced *Pcsk9* mRNA shifting toward heavier polysome fractions (Fig. 7p). These results implied that Lin28b/let-7 regulated translation of Pcsk9 rather than mRNA stability. In addition, IHC staining showed that Pcsk9 expression was substantially reduced in *FSP-Cre;Lin28b^(fl/fl)* mice compared to wild-type mice, demonstrating that Lin28b regulated Pcsk9 expression in

**Fig. 3 | Lin28b^high PDAC cells secret Wnt5a to induce Lin28b expression in CAFs. a** Venn diagram of genes upregulated in 15376T compared with Lin28b-KO 15376T and 14837T as determined by RNA-sequencing (RNA-seq). **b, c** The levels of Wnt5a in 15376T, Lin28b-KO 15376T, and 14837T were measured by real-time qPCR (**b**) and western blotting (**c**). Representative of *n* = 3 independent experiments (**c**). **d, e** The levels of WNT5A in PANC-1, LIN28B-KO PANC-1, and Mia Paca-2 were measured by real-time qPCR (**d**) and western blotting (**e**). Representative of *n* = 3 independent experiments (**e**). **f** Wnt5a levels in the supernatants of 15376T, Lin28b-KO 15376T, and 14837T were examined by ELISA. **g** WNT5A levels in the supernatants of PANC-1, LIN28B-KO PANC-1 and Mia Paca-2 were examined by ELISA. **h, i** 14837CAFs (**h**) or 15376CAFs (**i**) were cultured with 15376T-CM or Wnt5a-KO 15376T-CM for 6 days. Then, Lin28b levels were measured by western blotting. Lin28b levels in 15376T were included as a positive control. Representative of *n* = 3 independent experiments. **j–l** hCAFs were cultured with PANC-1-CM (**j**), WNT5A-KO PANC-1-CM (**j**), PANC03.27-CM (**k**), WNT5A-KO PANC03.27-CM (**k**), hPDAC1^#-CM (**l**), and WNT5A-KO hPDAC1^#-CM (**l**) for 6 days. Then, the levels of LIN28B were measured by western blotting. LIN28B levels in PANC-1 (**j**), PANC03.27 (**k**), and hPDAC1^# (**l**) were included as a positive control. Representative of *n* = 3 independent experiments. **m** Wnt5a levels in the supernatants of low (30–40%) and high (80–90%) density inoculated 15376T were examined by ELISA. **n–q** 14837CAFs (**n, o**) or 15376CAFs (**p, q**) were treated with 100 ng/ml or 200 ng/ml recombinant-Wnt5a(r-mWnt5a) for 6 days. Cells were collected for Western blotting at indicated time points. Lin28b levels in 15376T were included as a positive control. Representative of *n* = 3 independent experiments (**n–q**). **r, s** 14837CAFs (**r**) or 15376CAFs (**s**) were cultured with 14837T-CM or 15376T-CM in the presence or absence of 1 μg/ml Wnt5a neutralizing antibody (anti-Wnt5a) for 6 days. Then, cells were collected for Western blotting. Lin28b levels in 15376T were included as a positive control. Representative of *n* = 3 independent experiments (**r, s**). **t** Orthotopic PDAC tumors generated with 15376T or Wnt5a-KO 15376T were analyzed by Wnt5a, Lin28b and α-SMA immunofluorescence staining (*n* = 6 mice). Representative images are shown. Scale bar: 30 μm. Three biologically independent experiments were performed (**b, d, f, g, m**). Data are shown as mean ± s.d. *P*-values were determined by one-way ANOVA with Tukey's multiple comparison test (**b, d, f, g**) or two-tailed unpaired Student's t-tests (**m**).

CAFs in vivo (Fig. 7q, r). Taken together, our findings confirmed Pcsk9 was a direct target of Lin28b/let-7 in CAFs.

## Discussion

Here in our study, we found that a positive feedback loop between Wnt5a and Lin28b in pancreatic cancer cells (Fig. 8). Our data indicated that Wnt5a can serve as a signal transducer between tumor epithelium and CAFs. Wnt5a has been reported to be implicated in the pathogenesis of PDAC[22]. Importantly, Wnt5a expression is significantly elevated in the squamous subtype of PDAC[23] which exhibits the worst prognosis compared with the other subtypes[24–26]. To further study the role of Lin28b-Wnt5a axis in advanced human PDAC, we analyzed the expression profiles of the distinct molecular subtypes of human PDAC from proteogenomic analysis of 140 pancreatic cancers[27,28] and found that tumors of the squamous subtype exhibited elevated expression of Lin28b (Supplementary Fig. 5a). Moreover, expression of the Lin28b and Wnt5a were correlated with that of the squamous subtype signature (Supplementary Fig. 5b, c and Supplementary Data file 1), underscoring the association between Lin28b-Wnt5a axis and squamous subtype tumors.

Lin28b/let-7 pathway plays important roles in modulating mRNA stability and translational efficacy[9,10]. Previous studies showed that Lin28b stabilized Wnt5a mRNA in VSMCs (vascular smooth muscle cells) and Wnt5a/β-catenin was the positive regulator of Lin28b expression in PGCs (primordial germ cell)[29,30]. Here we used wild-type/mutant Lin28b plasmids and found that flag-Lin28b-WT(r), but not flag-Lin28b-MU(r), increased Wnt5a protein level which was suppressed by Lin28b knocking-out (Supplementary Fig. 6a). The rescue experiment proved that Wnt5a is a reliable target of Lin28b. In addition, Wnt5a was increased by let-7 sponge, indicating that Wnt5a was able to be regulated by Lin28b/let-7 pathway (Supplementary Fig. 6b). Moreover we also observed that Lin28b/let-7 pathway is able to regulate *Wnt5a* mRNA stability in PDACs (Supplementary Fig. 6c). What's more, Wnt5a secreted by tumor epithelium could also upregulate Lin28b in surrounding CAFs even in response to low glucose concentration (Supplementary Fig. 6d). In this way, CAFs could express high level of Lin28b which was supposed to express in embryonic stem cells and tumor cells only[15]. Furthermore, the role of Lin28b in CAFs was investigated by using orthotopic model in *FSP-Cre;Lin28b^{fl/fl}* mice. Although previous studies mainly focused on the Lin28b's function in tumors, the in vivo data from our study suggested that Lin28b could promote tumor growth in both cell-autonomous and non-cell-autonomous ways (Fig. 5o–r). We have used several human cell lines and found PANC-1 and PANC03.27 which expressed high level of LIN28B do dependency on LIN28B's function as reported before (Supplementary Fig. 7a, b). But we also observed the non-cell autonomous in the orthotopic mouse model which was generated in *FSP-*

*Cre;Lin28b^{fl/fl}* mice. The discrepancy between the findings in our study and previous reports may result from different tumor model we used. We also noticed that, although Lin28b did not affect proliferation of CAFs, Lin28b expression status is associated with secretion of cytokines by CAFs, such as C-X-C motif chemokine 5 (CXCL5) (Fig. 6c and Supplementary Fig. 3d, e). Since these cytokines are highly related with immunosuppression, we hypothesize that Lin28b-positive CAFs might also contribute to the immune-microenvironment in PDAC, which will be focused on in our further exploration.

Lin28b can be expressed in a variety of CAFs including aSMA^+ myofibroblasts and FAP^+ myofibroblasts which support tumor growth (Fig. 1i, j). Although previous study showed that aSMA^+ myofibroblasts restrain the tumor, but most of these myCAF are functional in inhibiting the metastasis of PDAC, as to tumor growth, it is still controversial. We cannot exclude the possibility that some subtype of aSMA^+ myofibroblasts might have the ability to promote tumor growth. For example, sc-RNAseq showed that several clusters of LRRC15^+ myofibroblasts which promote tumor growth also highly expressed aSMA[31]. Also, previous studies indicate that some subpopulation of myofibroblasts promotes the development and metastasis of PDAC[32,33].

Previous study showed that Lin28b could be induced by the loss of SIRT6[12]. We also found that SIRT6 expression level is negatively correlated with Lin28b in both human PDAC and murine PDAC (Supplementary Fig. 8a–d). Knocking-out Sirt6 did not induce Lin28b expression in 14837T cells even in the presence of 15376T conditioned medium or recombinant-Wnt5a treatment (Supplementary Fig. 8e). However, we found that Sirt6 loss in CAFs could induce Lin28b expression (Supplementary Fig. 8f). Since our data indicated that Wnt5a-Fzd4 pathway is essential for activation of Lin28b (Fig. 4c–f), we then tested the protein level of Fzd4 in 14837T and 14837CAFs and found Fzd4 which could be expressed in 14837CAFs but not 14837T (Supplementary Fig. 8e, f). Moreover, consistent with previous study[34], we found that Fzd4 expression was also induced by loss of SIRT6 (Supplementary Fig. 8e, f), suggesting that Fzd4-mediated SIRT6-induced Lin28b expression in CAFs.

CM from Lin28b^high or Lin28b^low tumors did not affect the survival and subtype switch of CAFs (Supplementary Fig. 9a–c and Supplementary Data file 2). In addition, although Lin28b depletion in CAFs did not affect tumor desmoplasia (Supplementary Fig. 9d, e), its expression is correlated with several cytokines secretion (Fig. 6c). Among these cytokines, we found that PCSK9 is essential for the growth-promoting effect of Lin28b-positive CAFs under low-glucose treatment. In addition, PCSK9 3'-UTR was identified as a direct target of Lin28b/let-7 (Fig. 7k, l). PCSK9 is closely associated with incidence and progression of several cancers[35]. It plays a central role in cholesterol homeostasis by increasing the levels of circulating low-density

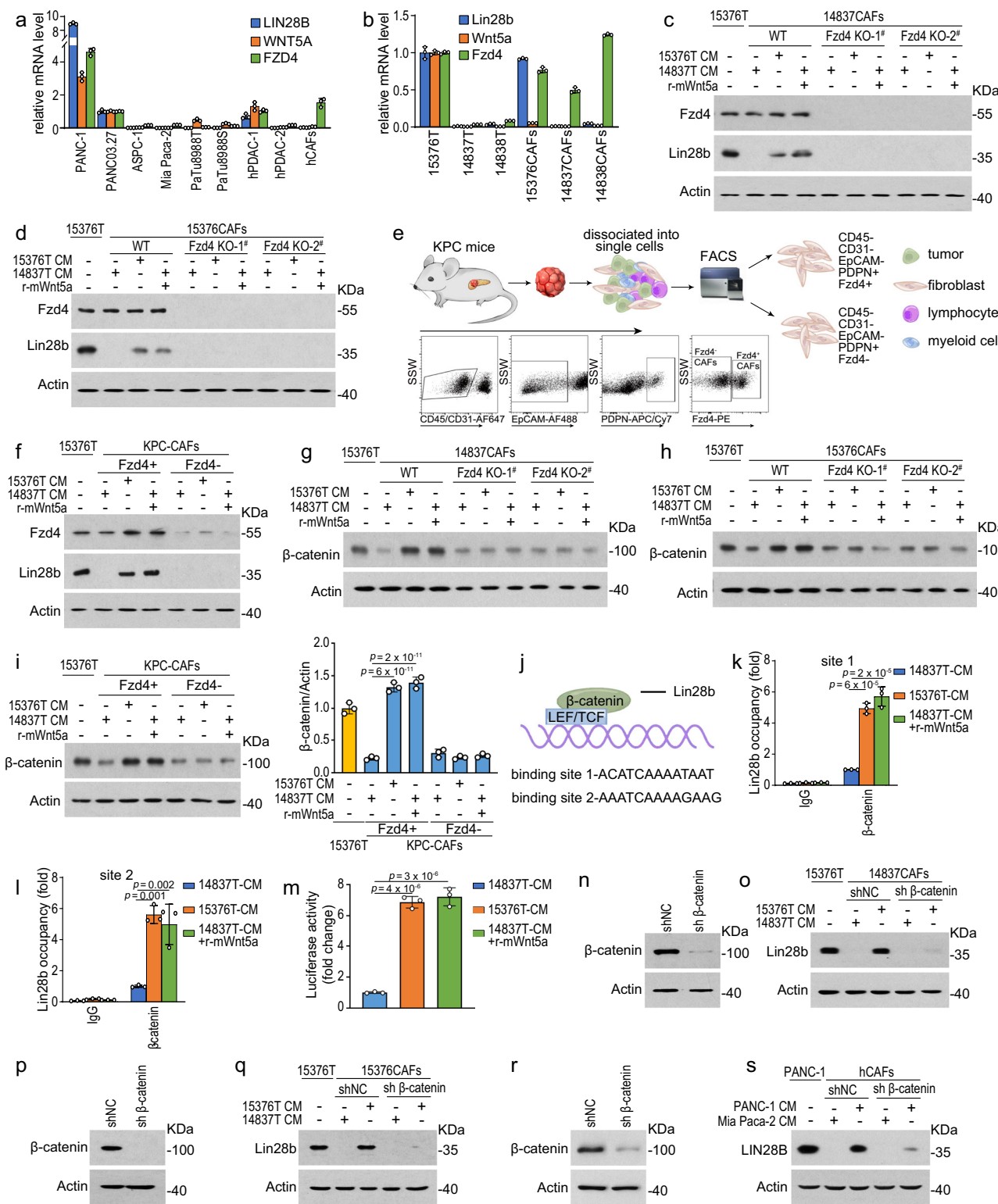

lipoprotein (LDL)-cholesterol (LDL-C)[36]. As a consequence, the increased plasma LDL-C levels could cause hypercholesterolemia that is associated with tumor cell growth[37]. It has been reported that LDL-C could potentiate colon cancer cell resistance to glucose-deprivation[38]. In our study, we found that recombinant Pcsk9 protein could decrease the protein level of Ldlr (Supplementary Fig. 10a). In addition, increase LDL-C level was able to sustain the PDAC survival in vitro in response to low-glucose treatment (Supplementary Fig. 10b). Overexpression of Ldlr diminished the effect of r-Pcsk9 protein on cell viability (Supplementary Fig. 10c, d). Moreover, PCSK9 inhibitor alirocumab blocked r-

Pcsk9's effect on tumor growth (Supplementary Fig. 10e), suggesting that Pcsk9 promotes tumor dependent on its binding with Ldlr. Recent studies uncovered more biological functions of PCSK9, including vascular inflammation, viral infections, or immune checkpoint modulation in cancer[36,39]. So, we cannot exclude the possibility that PCSK9 might promote tumor growth by other pathways besides inducing LDL-C. Currently, there are multiple therapeutic approaches targeting PCSK9 which have been evaluated, including monoclonal antibodies (mAbs), peptidic inhibitor, antisense oligonucleotides (ASOs), genome editing technique with CRISPR-CAS9, PCSK9-vaccine and small

**Fig. 4 | Wnt5a-Fzd4-β-catenin pathway plays a critical role in Lin28b induction. a, b** The mRNA levels of Lin28b, Wnt5a, and Fzd4 in indicated cells were measured by real-time qPCR. **c, d** 14837CAFs (**c**), Fzd4-KO 14837CAFs **c**, 15376CAFs and Fzd4-KO 15376CAFs (**d**) were cultured with 14837T-CM or 15376T-CM in the presence or absence of 100 ng/ml r-mWnt5a for 6 days. Then, cells were collected for Western blotting. Lin28b levels in 15376T were included as a positive control. Representative of $n = 3$ independent experiments. **e** Fzd4-positive (Fzd4[+]) and Fzd4-negative (Fzd4[-]) CAFs from KPC mice were sorted by flow cytometry (FACS). Flow plots showing the gating strategy for sorting Fzd4[+] and Fzd4[-] CAFs from KPC tumors. **f** Fzd4[+] CAFs and Fzd4[-] CAFs were cultured with 14837T-CM or 15376T-CM in the presence or absence of 100 ng/ml r-mWnt5a for 6 days. Then, cells were collected for Western blotting. Lin28b levels in 15376T were included as a positive control. Representative of $n = 3$ independent experiments. **g, h** 14837CAFs, Fzd4-KO 14837CAFs (**g**), 15376CAFs, and Fzd4-KO 15376CAFs (**h**) were cultured with 14837T-CM or 15376T-CM in the presence or absence of 100 ng/ml r-mWnt5a for 6 days. Then, the levels of β-catenin were measured by Western blotting. Representative of $n = 3$ independent experiments. **i** Fzd4[+] CAFs and Fzd4[-] CAFs were cultured with 14837T-CM or 15376T-CM in the presence or absence of 100 ng/ml r-mWnt5a for 6 days. Then, the levels of β-catenin were measured by Western blotting and quantification. Representative of $n = 3$ independent experiments. **j** A schematic illustration of the β-catenin binding sites in the Lin28b promoter. **k, l** 14837CAFs were cultured with 14837T-CM or 15376T-CM in the presence or absence of 100 ng/ml r-mWnt5a for 6 days. ChIP experiments were performed using anti-β-catenin antibody. k, binding site 1; l, binding site 2. **m** The relative luciferase activity was analyzed after 14837CAFs were transfected with TCF/LEF reporter and pRL-TK vector and then cultured with 14837T-CM or 15376T-CM in the presence or absence of 200 ng/ml r-mWnt5a for 3 days. **n–r** Western blotting was used to detect the knockdown efficiency of β-catenin in 14837CAFs (**n**) and 15376CAFs (**p**). Representative of $n = 2$ independent experiments. 14837CAFs and β-catenin-KD 14837CAFs (**o**), 15376CAFs and, and β-catenin-KD 15376CAFs (**q**) were cultured with 14837T-CM or 15376T-CM for 6 days. Then, cells were collected for Western blotting. Lin28b levels in 15376T were included as a positive control. Representative of $n = 3$ independent experiments (**o, q**). **r, s** Western blotting was used to detect the knockdown efficiency of β-catenin in hCAFs (**r**). Representative of $n = 2$ independent experiments. hCAFs and β-catenin-KD hCAFs were cultured with Mia Paca-2-CM or PANC-1-CM for 6 days. Then, cells were collected for Western blotting (**s**). Lin28b levels in PANC-1 were included as a positive control. Representative of $n = 3$ independent experiments (**s**). Three biologically independent experiments were performed (**a, b, i, k–m**). Data are shown as mean ± s.d. $P$-values were determined by one-way ANOVA with Tukey's multiple comparison test (**a, b, i, k–m**).

molecule inhibitors of PCSK9[35]. Our findings may implicate PCSK9 inhibition as a means for pancreatic cancer treatment.

# Methods

## Ethics approval and consent to participate
Mice (male/female) received standard chow diet freely and were housed in conditions of 12:12 h dark:light cycle, $22 \pm 1\,°C$ ambient temperature, and $50 \pm 10\%$ humidity at Peking University Health Science Center animal facility. All animal experiments were performed in accordance with a protocol approved by the Department of Laboratory Animal Science of Peking University Health Science Center and supervised by the institutional review board of Peking University.

Human PDAC cell line (hPDAC), CAFs (hCAFs), and human TMA samples were collected from PDAC patients (male/female) aged 34–82 years diagnosed with PDAC at the Department of Pancreatic Surgery, Huashan Hospital. All manipulations about human patient samples were approved by the Research Ethics Committee of Huashan Hospital, Fudan University. All patients provided written informed consent before enrollment and have consented to the data reporting provided in Supplementary Data file 3.

## Transgenic mice cell lines
Primary mouse PDAC lines 14837T, 14838T, and 15376T were isolated as described[40] from genetically engineered C57BL/6 mice (*TetO-Lox-Stop-Lox-Kras*$^{G12D}$ (*tetO_LKras*$^{G12D}$), *ROSA26-LSL-rtTA-IRES-GFP* (*ROSA_rtTA*), *p48-Cre*, and Trp53$^{L/+}$)[41], and maintained in DMEM medium containing 10% FBS, 1% penicillin/streptomycin, and 1 μg/mL doxycycline.

## Mouse models
Mice (male/female) for fibroblast deletion of Lin28b were generated by breeding *Lin28b*$^{fl/fl}$ mice (Cyagen Biosciences, Suzhou, China) with *FSP-Cre* mice (Model Organisms Centre, Shanghai, China). All genotyping was done by PCR. Male or female mice were randomly selected in all animal experiments.

## Cell culture
The human pancreatic cancer cell lines PANC-1, PANC03.27, ASPC-1, Mia Paca-2, PaTu8988T, and PaTu8988S were grown in DMEM with 10% FBS and 1% penicillin/streptomycin in a $37\,°C$ incubator with a humidified, 5% $CO_2$ atmosphere. The human pancreatic cancer cell lines were purchased from American Type Culture Collection (ATCC).

The mCAFs were generated from C57BL/6J mice harboring mouse PDAC. These animals were pre-treated with a doxycycline diet and kept on a doxycycline regimen for the duration of the experiment, and were injected with $10^5$ 14837T, 14838T, or 15376T cells into the pancreas. Pancreatic tumors were resected at 2 weeks, mechanically minced and digested in collagenase and dispase. Cells were plated in cell culture dishes in DMEM with 10% FBS in the absence of doxycycline to limit the growth of iKRAS murine PDAC cells. The mCAFs were immortalized by infection with SV40 (Addgene, plasmid no. 22298) retroviruses.

The hCAFs were prepared by the outgrowth method. Fresh tissue was obtained from residual pancreatic adenocarcinoma specimens of patients undergoing primary surgical resection. It was digested in collagenase and dispase and mechanically minced. Primary human pancreatic CAFs were isolated by differential trypsinization and immortalized by infection with SV40 (Addgene, plasmid no. 22298) retroviruses. Cells were kept in DMEM supplemented with 10% FBS.

CAFs were verified by measuring α-SMA expression.

## Direct contact co-culture and FACS
Primary mouse PDAC cells were direct co-cultured with CAFs (1:1) for 6 days. Primary mouse PDAC cells from genetically engineered mice expressed green fluorescence, while CAFs from WT mice did not. Thus, PDAC cells and CAFs could be directly selected by flow cytometric cell sorting (FACS). Before cell sorting, PDAC cells and CAFs were resuspended in PBS containing 1% FBS. FACS was performed on single-cell suspensions using flow cytometer (BD Influx). The purity of the sorted populations was verified by flow cytometry.

## Flow cytometry and sorting of KPC CAFs
High-resolution ultrasound imaging was used to identify a dominant mass that met enrollment criteria. For sorting of Fzd4[+] CAFs and Fzd4[-] CAFs, tumor tissues from KPC (*Kras*$^{LSL-G12D/+}$; *Trp53*$^{R172H/+}$; *Pdx1*-Cre) mice were minced and enzymatically digested in PBS supplemented with 5% FBS, 1 mg/ml Type VIII Collagenase (Sigma, #C2139), 2 mg/ml Dispase II (Sigma, #4942078001), 1 mg/ml Trypsin Inhibitor (Sigma, #T6522) and 1 unit/ml DNase I (NEB, # M0303S) for 45 min at $37\,°C$ with agitation. Cell digestion of both tumor and adjacent normal samples were strained through 100 μm cell strainer and resuspended in PBS supplemented with 0.5% FBS. Cells were stained with anti-Fzd4 (R&D Systems, #MAB194), and rabbit anti-rat IgG-PE (Solarbio, #K0032R-PE) was used as the secondary antibody. Cells were subsequently stained with anti-CD45-AlexaFluor 647 (BioLegend, #103124), anti-CD326 (EpCAM)-AlexaFluor 488 (BioLegend, #118212), anti-CD31-AlexaFluor 647 (BioLegend, #102416) and anti-PDPN-APC/Cy7 (BioLegend, #127418). Cells were sorted on the FACSAria$^{TM}$ II cell sorter (BD) for CD45/CD31/EpCAM[-] PDPN[+] Fzd4[+] and CD45/CD31/EpCAM[-] PDPN[+] Fzd4[-] cell populations.

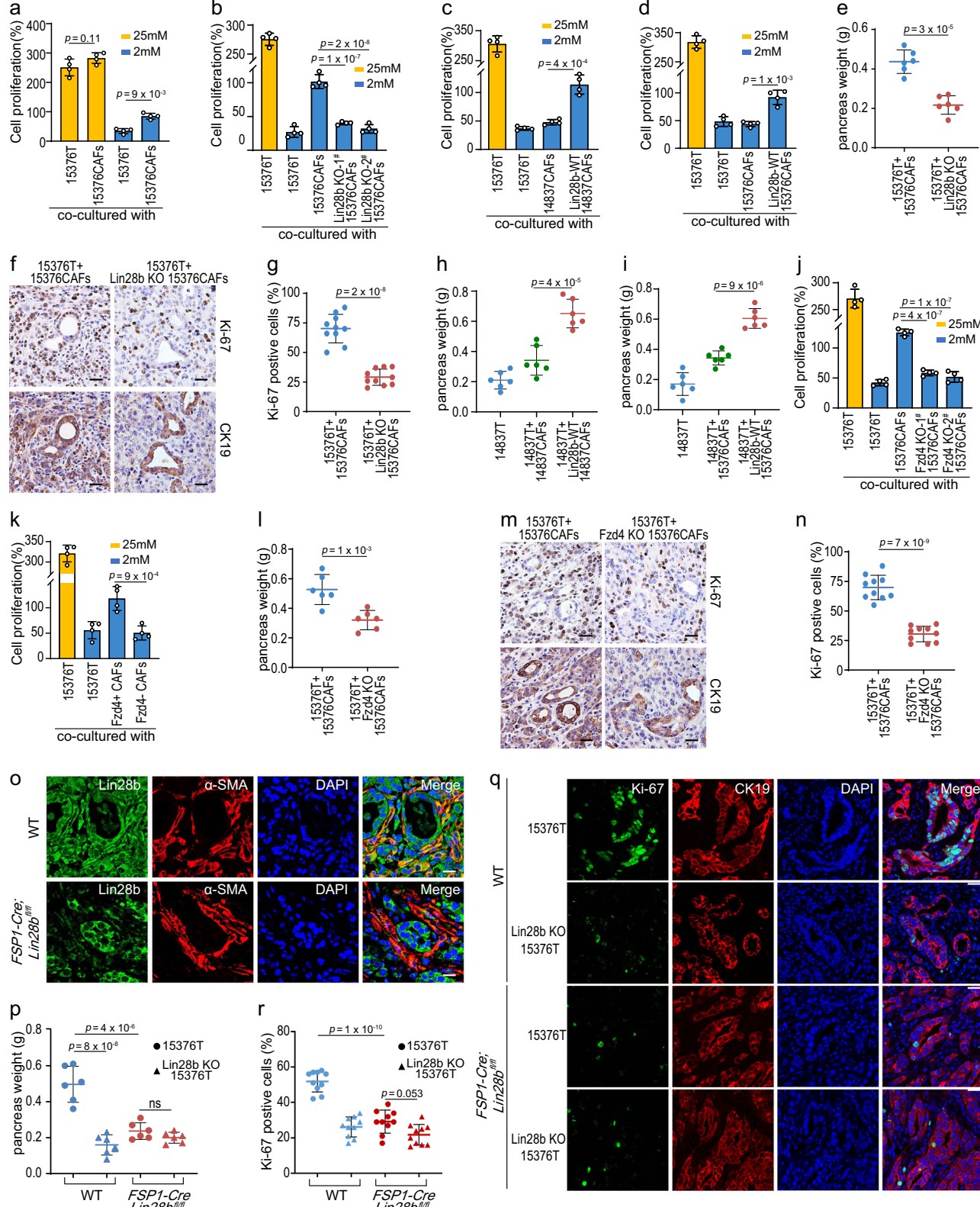

**Indirect co-culture experiments (Transwell)**
Co-culture experiments were performed by seeding CAFs cells ($1 \times 10^5$) in the lower chamber and PDAC cells ($1 \times 10^5$) in the upper chamber of a 6-well transwell apparatus with 0.4 μm pore size (Corning Incorporated, NY, USA). The co-cultured cells were passaged once they grew to 90% of confluence.

**Conditioned medium (CM)**
Conditioned medium (CM) was generated by adding fresh medium to cells at >50% confluence and was collected after 2 days of culture. For Lin28b induction experiments, CM and reagents for CAFs culture was replaced daily. For size cut-off experiments CM was filtered through 3-kDa cutoff columns (EMD Millipore, UFC900308). Concentrated

**Fig. 5 | Lin28b-positive CAFs promote growth of pancreatic cancer.** Before the experiment started, CAFs were cultured with 15376T-CM for 6 days to induce Lin28b expression. CM for CAFs culture was replaced daily (**a**, **b**, **e–h**, **k–p**). **a** 15376T were co-cultured with or without 15376CAFs in transwell chambers and treated with complete media (25 mM glucose) or glucose-limited medium (2 mM glucose) for 2 days. The cells were counted to calculate the cell proliferation. **b** Tumors were co-cultured with 15376CAFs or Lin28b-KO 15376CAFs in transwell chambers for 2 days. The cells were counted to calculate the cell proliferation. **c**, **d** Lin28b-WT was stable expressed in 14837CAFs (**c**) and 15376CAFs (**d**). Tumors were co-cultured with 14837CAFs (**c**), Lin28b-WT-expressing 14837CAFs (**c**), 15376CAFs (**d**) or Lin28b-WT-expressing 15376CAFs (**d**) in transwell chambers for 2 days. The cells were counted to calculate the cell proliferation. **e–g** 15376T was orthotopically co-injected with 15376CAFs or Lin28b-KO 15376CAFs into C57BL/6J mice ($n = 6$ mice). After 1 week, the pancreas was weighed (**e**) and analyzed by Ki-67 and CK19 IHC staining (**f**). Scale bar: 30 μM. The proportion of ki67-positive (**g**) cell was shown ($n = 10$ views per group). Data are shown as mean ± s.d. **h**, **i** 14837T was orthotopically co-injected with or without 14837CAFs (**h**), Lin28b-WT-expressing 14837CAFs (**h**), 15376CAFs (**i**) or Lin28b-WT-expressing 15376CAFs (**i**) into C57BL/6J mice ($n = 6$ mice). After 1 week, the pancreas was weighed. **j** Tumors were co-cultured with 15376CAFs or Fzd4-KO 15376CAFs in transwell chambers for 2 days. The cells were counted to calculate the cell proliferation. **k** Tumors were co-cultured with Fzd4⁺ CAFs or Fzd4⁻ CAFs in transwell chambers for 2 days. The cells were counted to calculate the cell proliferation. **l–n** 15376T was orthotopically co-injected with 15376CAFs or Fzd4-KO 15376CAFs into C57BL/6J mice ($n = 6$ mice). After 1 week, the pancreas was weighed (**l**) and analyzed by Ki-67 and CK19 staining (**m**). Scale bar: 30 μM. The proportion of ki67-positive (**n**) cell was shown ($n = 10$ views per group). **o** 15376T cells were orthotopically injected into WT or *FSP-Cre;Lin28b^{fl/fl}* mice ($n = 6$ mice). After 2 weeks, the tumors were analyzed by Lin28b and α-SMA immunofluorescence staining. Representative images are shown. Scale bar: 30 μM. **p–r** 15376T or Lin28b-KO 15376T were orthotopically injected into WT or *FSP-Cre;Lin28b^{fl/fl}* mice ($n = 6$ mice). After 2 weeks, the pancreas was weighed (**p**) and analyzed by Ki-67 and CK19 immunofluorescence staining (**q**). Scale bar: 30 μM. The proportion of ki67-positive (**r**) cell was shown ($n = 10$ views per group). Four biologically independent experiments were performed (**a–d**, **j**, **k**). Data are shown as mean ± s.d. *P*-value were determined by one-way ANOVA with Tukey's multiple comparison test (**a–d**, **h–k**, **p**, **r**) or two-tailed unpaired Student's t-tests (**e**, **g**, **l**, **n**).

(>3 kDa) medium was resuspended in a DMEM volume matching the initial medium volume. Boiled medium experiments were performed by heating CM at 100 °C for 15 min followed by filtration at 0.45 μm to remove precipitate. Freeze-thaw medium was treated by 3 consecutive cycles of 15 min at −80 °C followed by 15 min at 60 °C and then filtered to remove precipitate.

## Cell number
Cells were plated in complete medium at 50% confluency. Once the cells were attached, medium was replaced with starvation medium or CM. Two days later, cells were trypsinized, re-suspended in their own medium, diluted in trypan blue (Thermo-scientific 15250061) and counted using a hemocytometer.

## Xenografts
$1 \times 10^5$ PDAC cells alone or mixed with CAFs at a ratio of 1:3 were implanted into the pancreas of 6-weeks-old C57BL/6J mice (male/female). Briefly, an incision was made on the flank, above the spleen. The spleen was identified and gently pulled out through the incision to expose the pancreas. Then 20 μl of cell suspension was injected into the tail of the pancreas. The spleen and pancreas were carefully reintroduced into the animal and the peritoneum sutured. Mice were fed with doxy water (doxycycline 2 g/l in sucrose 20 g/l). Two weeks after inoculation, tumors were resected and weighed. Maximal tumor burden (10% of body weight) and maximal tumor size (2.0 cm) allowed by the ethics committee were not exceeded.

## Antibodies and reagents
The antibodies used were anti-Lin28B (Abcam, #ab191881, 1:1000; abcepta, #AP1485C, 1:1000), anti-actin (Santa Cruz, #sc-7210, 1:1000), anti-Wnt5a (Abcam, #ab229200, 1:500; R&D, #MAB645, 1:200), anti-Wnt10a (ABclonal, #A15602, 1:1000; biodragon, #BD-PNO283, 1:1000), anti-β-catenin (Abcam, #ab32572, 1:10,000), anti-Pcsk9 (Abcam, #ab185194, 1:1000), anti-PGRN (Abcam, #ab187070, 1:1000), and anti-Ldlr (Abcam, #ab52818, 1:500) for WB; anti-Ki67 (Abcam, #ab16667, 0.03 μg/ml), anti-α-SMA (Gene Tech, #GM085129, 1:100), anti-CK19 (Proteintech, #10712-1-AP, 0.5 μg/ml), anti-Lin28B (Abcam, #ab191881, 1:50; abcepta, #AP1485C, 1:50) and anti-Pcsk9 (Abcam, #ab185194, 1:100) for IF and IHC; anti-Fzd4 (R&D Systems, #MAB194, 1:100), rabbit anti-rat IgG-PE (Solarbio, #K0032R-PE, 1:100), anti-CD45-AlexaFluor 647 (BioLegend, #103124, 1:200), anti-CD326 (Ep-CAM)-AlexaFluor 488 (BioLegend, #118212, 1:25), anti-CD31-AlexaFluor 647 (BioLegend, #102416, 1:200) and anti-PDPN-APC/Cy7 (BioLegend, #127418, 1:200) for FACS. Recombinant protein of Wnt5a (#APP549Mu01) was purchased from Cloud-Clone. Recombinant protein of Pcsk9 (#50251-M08H) and PGRN (#50396-M08H) was purchased from SinoBiological.

Doxycycline (#S5159) was purchased from Selleck. Actinomycin D (#HY-17559) and alirocumab (#HY-P9928A) were purchased from MedChemExpress (MCE). Type VIII Collagenase (#C2139) was purchased from Sigma. Dispase II (#4942078001) was purchased from Sigma. Trypsin Inhibitor (#T6522) was purchased from Sigma. DNase I (# M0303S) was purchased from NEB.

## Gene expression analysis by real-time PCR (qPCR)
Total RNA was extracted with the TRIzol (Applygen) as described by the manufacturer. For cDNA synthesis, 1 μg of total RNA was reverse-transcribed using the 1st Strand cDNA Synthesis Kit (gDNA digester plus) (Yeasen). Real-time PCR was run in duplicate using Taq Pro Universal SYBR qPCR Master Mix (Vazyme), following the manufacturer's instructions. Relative gene expression was determined by real-time PCR using an Applied Biosystems 7500 Real-Time PCR System (Applied Biosystems, CA, USA) according to the manufacturer's recommended protocol. Each analysis was performed in triplicate. Sequences for qPCR primers are as follows: actin(mouse): ctaaggccaa ccgtgaaaag, accagaggcatacagggaca; actin(human): ccaaccgcgagaaga tga, ccagaggcgtacagggatag; 5sRNA: ggccataccaccctgaacgc, cagcacc cggtattcccagg; Lin28b(mouse): gccttgagtcaatacgggtaac, agggtcttccct ttaggtcttc; LIN28B(human): catctccatgataaaccgagagg, gttacccgtattga ctcaaggc; Wnt5a(mouse): caactggcaggactttctcaa, catctccgatgccggaa ct; WNT5A(human): tcgactatggctaccgctttg, cactctcgtaggagcccttg; Wnt10a(mouse): gctcaacgccaacacagtg, cgaaaacctcggctgaagatg; Fzd4 (mouse): tgccagaacctcggctaca, atgagcggcgtgaaagttgt; FZD4(human): gtctttcagtcaagagacgctg, gttgtggtcgttctgtgggt; Pcsk9(mouse): gagaccc agaggctacagatt, aatgtactccacatggggcaa; Clec3b(mouse): ccttcaccc aaccgaagacc, tcgttctctagctctgactgc; Gsn(mouse): atggctccgtaccgctctt, gcctcagacacccgactttt; ptx3(mouse): cctgcgatcctgctttgtg, ggtgggat gaagtccattgtc; Ly6c1(mouse): gcagtgctacgagtgctatgg, actgacgggtcttta gtttcctt; Efemp1(mouse): gcgctggtcaagtcacagta, aagcatctgggaca atgtcac; Ly6a(mouse): aggaggcagcagttattgtgg, cgttgaccttagtacccagga; Figf(mouse): ttgagcgatcatcccggtc, gcgtgagtccatactggcaag; Tnfaip6(m ouse): gggattcaagaacgggatcttt, tcaaattcacatacggccttgg; Ifi27l2a(m ouse): gcttgttgggaaccctgtttg, ggatggcatttgttgatgtggag; Dpt(mouse): tggatggtgaatcttaaccgc, tcagagccttccttcttgcta; Adm(mouse): caccctg atgttattgggttca, ttagcgcccacttattccact; Plpp3(mouse): tcgtccctgagag taagaacg, tgcttgtctcgatgatgaggaa; Tnxb(mouse): tccgtgtagactcagcaaa gg, ccccacgataagagacagcg; Cxcl12(mouse): tgcatcagtgacggtaaacca, ttc ttcagccgtgcaacaatc; Gstm1(mouse): atactgggatactggaacgtcc, agtcaggg ttgtaacagagcat; C4b(mouse): acttcagcagcttagtcaggg, gtcctttgtttcag gggacag; Ogn(mouse): accataacgacctggaatcgt, aacgagtgtcattagcctt gc; C3(mouse): ccagctccccattagctctg, gcacttgcctcttttaggaagtc; Pcolce2 (mouse): tgtggcggcattcttaccg, ccctcaggaactgtgattttcca; Cxcl1(mouse): ctgggattcacctcaagaacatc, cagggtcaaggcaagcctc; Col14a1(mouse): tttg

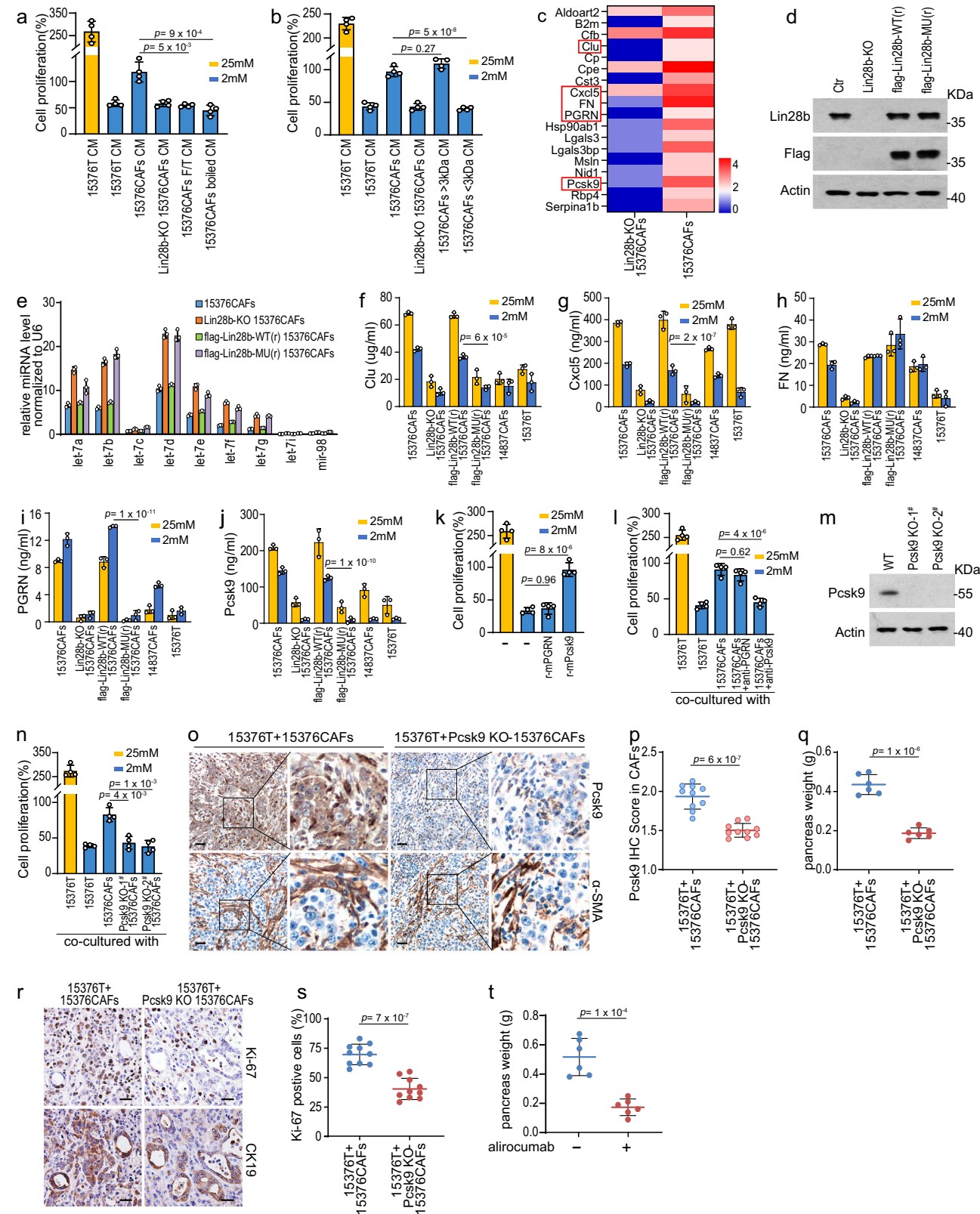

gcggctgcttgtttc, cgcttttgttgcagtgttctg; Svep1(mouse): actaggcttacc
tgtcaaggaa, ggtggagaaatgatgacgcc; Adamts5(mouse): ggagcgaggccatt
tacaac, cgtagacaaggtagcccactttt; Hp(mouse): gctatgttggagcacttggttc,
cacccattgcttctcgtcgtt; Has1(mouse): ggcgagcactcacgatcatc, aggagtc
catagcgatctgaag; Scara3(mouse): tgacagggatgtactgtgtgt, tgcaaagatag
gttcttctggc; Ifi205(mouse): aagatcaaggcatctgggaaag, cctctgggaatgtt

ctggttc; Dpep1(mouse): gcacaacgacttgccttgg, atgcggtgtatcacatccatc;
Sfrp4(mouse): agaaggtccatacagtgggaag, gttactgcgactggtgcga; Prss23
(mouse): ggtgagtccctacaccgttc, ggcgtcgaagtctgccttag; Ackr3(mouse):
agcctggcaactactctgaca, gaagcacgttcttgttaggca; Htra3(mouse): gtaag
ctccaccaggaatcaaa, gccagagggttgtgtgaacta; Apoe(mouse): ctgacaggat
gcctagccg, cgcaggtaatcccagaagc; Ccl7(mouse): gctgctttcagcatccaagtg,

**Fig. 6 | Pcsk9 secreted by CAFs promotes PDAC growth.** Before the experiment started, CAFs were cultured with 15376T-CM for 6 days to induce Lin28b expression. CM for CAFs culture was replaced daily. **a** The ability of CAFs-CM to increase PDAC proliferation was abolished after boiling at 100 °C for 15 min as well as after three consecutive freeze (−80 °C, 10 min)-thaw (60 °C, 10 min) cycles. **b** The factor secreted by CAFs that increases PDAC proliferation was retained in the >3-kDa fraction of CAFs-CM. **c** CAFs were cultured with 15376T-CM for 6 days and then they were cultured with FBS-free 15376T-CM for 24 h. Then, CM from 15376CAFs or Lin28b-KO 15376CAFs were harvested for quantitative secretomics analysis. A heat map shows cytokines which are downregulated in Lin28b-KO 15376CAFs-CM. **d, e** CRISPR/Cas9-resistent flag-Lin28b-WT(r) and flag-Lin28b-MU(r) were overexpressed in Lin28b-KO CAFs. Western blotting was then performed to determine Lin28b protein levels (**d**). let-7 (let-7a, let-7b, let-7c, let-7d, let-7e, let-7f, let-7g, let-7i, mir-98) levels were measured by real-time qPCR (**e**). Representative of $n = 3$ independent experiments (**d**). **f–j** The levels of Clu (**f**), Cxcl5 (**g**), FN (**h**), PGRN (**i**), and Pcsk9 (**j**) in the supernatants of indicated cells were examined by ELISA. **k** 15376T were treated with 100 ng/ml recombinant PGRN (r-mPGRN) or 100 ng/ml recombinant Pcsk9 (r-mPcsk9) under low glucose (2 mM) for 2 days. The cells were counted to calculate the cell proliferation. **l** 15376T were co-cultured with 15376CAFs in transwell chambers, and treated with or without 1 µg/ml neutralizing antibody (anti-Pcsk9 or anti-PGRN) under low glucose (2 mM) for 2 days. The cells were counted to calculate the cell proliferation. **m, n** Western blotting was used to detect the knocking-out efficiency of Pcsk9 in 15376CAFs (**m**). Representative of $n = 2$ independent experiments. 15376T were co-cultured with 15376CAFs or Pcsk9-KO 15376CAFs in transwell chambers for 2 days. The cells were counted to calculate the cell proliferation (**n**). **o–s** 15376T was orthotopically co-injected with 15376CAFs or Pcsk9-KO 15376CAFs into C57BL/6 J mice and the tumors were harvested after 1 week ($n = 6$ mice). IHC was performed with Pcsk9 and α-SMA antibodies (**o**) and Pcsk9 IHC scores in CAFs were plotted ($n = 10$ views per group) (**p**). The pancreas was weighed (**q**) and analyzed by Ki-67 and CK19 IHC staining (**r**). The proportion of ki67-positive (**s**) cell was shown ($n = 10$ views per group). Scale bar: 30 µM. (**t**) 15376T were orthotopically injected into C57BL/6 J mice. About 200 µg anti-Pcsk9 monoclonal antibodies (alirocumab) were intraperitoneally injected on days 3, 5, 8, and 11. After 2 weeks, the pancreas was weighed ($n = 6$ mice). Three biologically independent experiments were performed (**e, f–j, t**). Four biologically independent experiments were performed (**a, b, k, l**). Data are shown as mean ± s.d. *P*-values were determined by one-way ANOVA with Tukey's multiple comparison test (**a, b, e–l, n**) or two-tailed unpaired Student's t-tests (**p, q, s, t**).

ccagggacaccgactactg; Il6(mouse): tagtccttcctaccccaatttcc, ttggtcct tagccactccttc; Sfrp2(mouse): cgtgggctcttcctcttcg, atgttctggtactcg atgccg; Pla1a(mouse): ggttgtggggaccacttttatg, caccttgaggttggtgcct; Scara5(mouse): tgggaagctaggggctacg, cggcaacattcagctctctct; Sned1 (mouse): atacagtgcgtgatctgatgc, ggtgatgatgtagaggtgtgc; Spp1 (mouse): agcaagaaactcttccaagcaa, gtgagattcgtcagattcatccg; Serpine2 (mouse): cacatgggatcgcgtccatc, cagcactttaccaactccgttta; Cxcl14 (mouse): gaagatggttatcgtcaccacc, cgttccaggcattgtaccact; Crlf1 (mouse): ctccctgcaagctacctgc, agggtggaggtgttaaggagg; Igfbp3 (mouse): ccaggaaacatcagtgagtcc, ggatggaacttggaatcggtca; Acta2 (mouse): gtcccagacatcagggagtaa, tcggtacttcagcgtcagga; Tagln (mouse): caacaagggtccatcctacgg, atctgggcggcctacatca; Thy1(mouse): tgctctcagtcttgcaggtg, tggatggagttatccttggtgtt; Col8a1(mouse): actct gtcagactcattcaggc, caaaggcatgtgagggacttg; Cthrc1(mouse): cagtt gtccgcaccgatca, ggtccttgtagacacattccatt; Sfrp1(mouse): caacgtgggcta caagaagat, ggccagtagaagccgaagaac; Tnc(mouse): acggctaccacaga agctg, atggctgttgttgctatggca; Sparcl1(mouse): ggcaatcccgacaagtacaag, tggttttctatgtctgctgtagc; Col15a1(mouse): cccagggaagaatggagaagt, ccagagccttcaatctcaaatcc; Col12a1(mouse): aagttgacccaccttccgac, ggt ccactgttattctgtaaccc; Tgfb1(mouse): ctcccgtggcttctagtgc, gccttagtttgag acaggatctg; Col1a1(mouse): gctcctcttaggggccact, ccacgtctcaccattgg gg; Sdc1(mouse): ctttgtcacggcagacacctt, gacagaggtaaaagcagtctcg; Cilp(mouse): atggcagcaatcaagacttgg, aggctggactcttctcactga; H19 (mouse): gaacagaagcattctaggctgg, ttctaagtgaattacggtgggtg; Thbs2 (mouse): ctggggcatagggccaagag, gcttgacaatcctgttgagatca.

For microRNA, it was extracted with MiPure Cell/Tissue miRNA Kit (Vazyme, China) and reverse transcribed into cDNA using miRNA 1st Strand cDNA Synthesis Kit (by stem-loop) (Vazyme, China), according to the manufacturer's instructions. Real-time PCR was run in duplicate using miRNA Universal SYBR qPCR Master Mix (Vazyme). Sequences for stem-loop primers and qPCR primers are designed according to the kit instructions.

### Genomic DNA isolation and genotype identification
genomic DNA of tumors and CAFs were extracted with TIANamp Genomic DNA Kit (TIANGEN, #DP304-02) as described by the manufacturer. The target regions of DNA were PCR amplified and separated by 2% agarose gel electrophoresis. Sequences for PCR primers are as follows: tetO_LKras$^{G12D}$ (about 600 bp): atgactgagtataaaacttgtggtgg, tcacataactgtacaccttgtccttgac; internal control (about 300 bp): caatggtaggctcactctgggagatgata, aacacacactggcaggactggctagg; p48-Cre (about 400 bp): cctggaaaatgcttctgtccg, cagggtgttataagcaatccc.

### Western Blotting
Cells were harvested and then lysed in lysis buffer (1% NP-40, 150 mM NaCl, 50 mM Tris, pH 7.5, 5 mM EDTA, 0.05% SDS and 1% EDTA-free protease and phosphatase inhibitor cocktails; Roche Applied Science) on ice for 25 min. After centrifugation at 12,000 × g. and 4 °C for 15 min, the protein concentration was measured with Coomassie brilliant blue (CBB). Equal amounts of proteins (20–40 µg) were size fractionated by 6–15% SDS–polyacrylamide gel electrophoresis. After transfer to nitrocellulose membranes, the membranes were blocked with milk at room temperature for 1 h and incubated with primary antibody at 4 °C overnight, and the secondary antibody at 4 °C for 1 h. Protein bands were analyzed using Image-Pro Plus 6.0 (Media Cybernetics Inc.). In this work, the target proteins were normalized to actin.

### ELISA
Cells were cultured in DMEM with 10% FBS until 80% of confluency, then washed with PBS and cultured in fresh serum-free media. Supernatants were harvested 24 h later and used for subsequent ELISA assay. The mouse Wnt5a ELISA kits (#MM-45764M2), human WNT5A ELISA kits (#MM-50715H2), mouse Clu ELISA kits (#MM-0923M2), mouse Ccl2 ELISA kits (#MM-0723M2), mouse Cxcl1 ELISA kits (#MM-43835M2), mouse Cxcl5 ELISA kits (#MM-0499M2) were purchased from MEIMIAN; the mouse Wnt10a ELISA kits (#AE6516A), human WNT10A ELISA kits (#AE6515A), mouse FN ELISA kits (#ER12667), mouse PGRN ELISA kits (#ER12449), mouse Pcsk9 ELISA kits (#ER11249) was purchased from Zancheng Technology. All experiments were performed according to the manufacturer's instructions.

### Generation of CRISPR/Cas9 KO cell lines
The Cas9 knockout cell lines were generated using CRISPR-Cas9 methods. We used the lentiCRISPRv2 puro vector purchased from Addgene (#98290). The sgRNA was designed by online software (http://crispr.mit.edu). pMDLg/pRRE (#12251), pRSV-REV (#12253), and pCMV-VSV-G (#8454) vectors were used together with lentiCRISPRv2 to generate lentiviruses. These plasmids were transfected into 293T cells and after 2 days the virus-containing supernatant of these cultures was collected and concentrated. Cells were infected with the lentiviruses supplemented with 4 µg/ml Polybrene. The puromycin selection (2.5 µg/mL) started 2 days after the infection and was continued for a week to generate stable cell pools. The sgRNA sequences are as follows: Lin28b-1#(mouse): sense: CACCGCATCTCCATGAT AAGTCGAG; antisense: AAACCTCGACTTATCATGGAGATGC; Lin28b-2#(mouse): sense: CACCGATACGGGTAACAGGCCCAGG; antisense: AAACCCTGGGCCTGTTACCCGTATC; LIN28B-1#(human): sense: CACC GACATCGACTGGAATATCCAA; antisense: AAACTTGGATATTCCAGTC GATGTC; LIN28B-2#(human): sense: CACCGATTCCTCCTCTGCCGGC TC; antisense: AAACGAGCCGGCAGAGGAGGAATC; Wnt5a-1#(mouse): sense: CACCGTGGAATATTAAGCCCGGGAG; antisense: AAACCTCCC GGGCTTAATATTCCAC; Wnt5a-2#(mouse): sense: CACCGACCAGTT

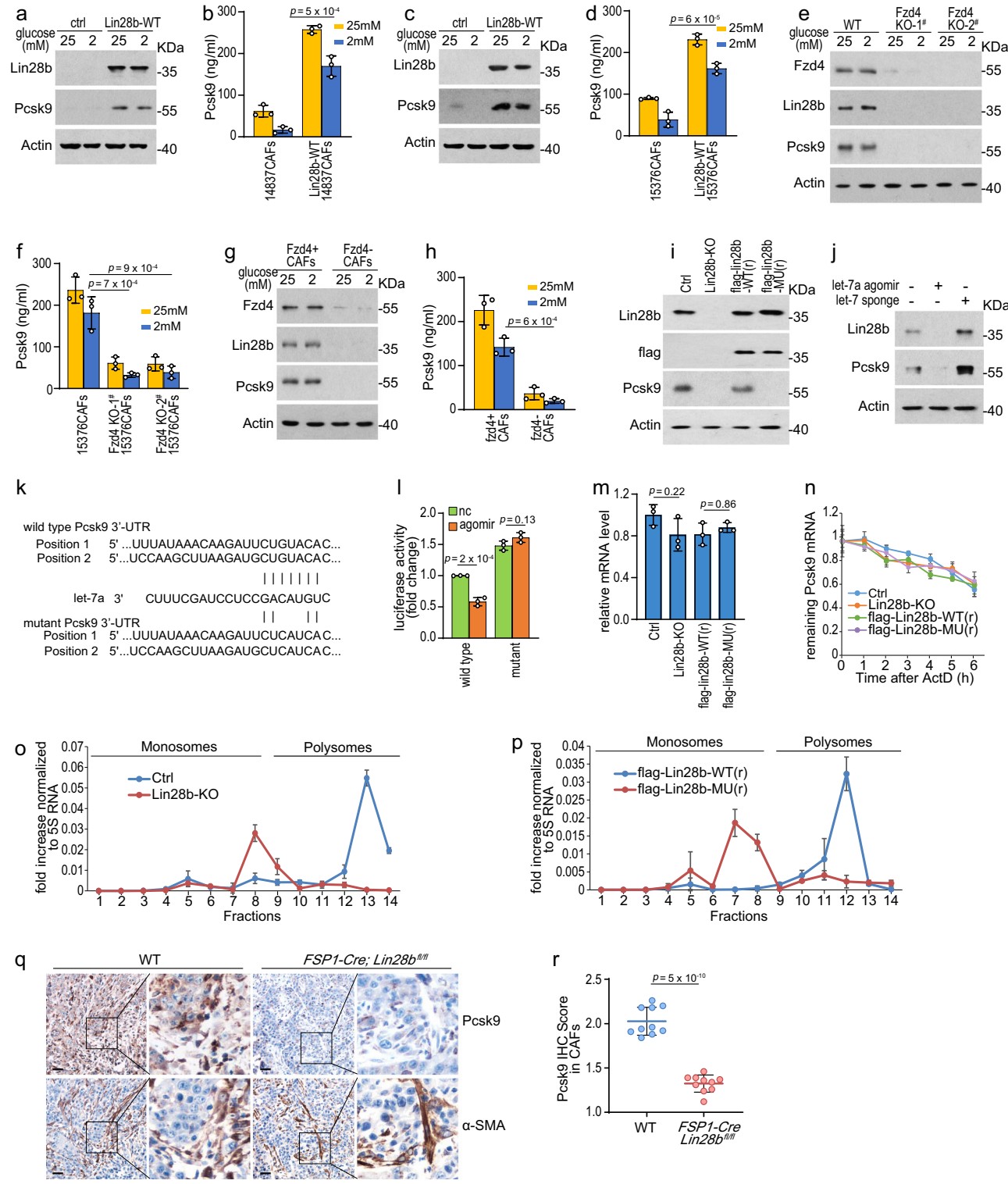

CCGGCATCGGAGA; antisense: AAACTCTCCGATGCCGGAACTGGTC; WNT5A-1#(human): sense: CACCGAGTATCAATTCCGACATCGA; antisense: AAACTCGATGTCGGAATTGATACTC; WNT5A-2#(human): sense: CACCGATCGGAGAAGGCGCGAAGAC; antisense: AAACGTCTT CGCGCCTTCTCCGATC; Wnt10a-1#(mouse): sense: CACCGTTGGCG TTGAGCACGGGCTC; antisense: AAACGAGCCCGTGCTCAACGCCAAC; Wnt10a-2#(mouse): sense: CACCGGCACTCATGGATGGCGATC; antisense: AAACGATCGCCATCCATGAGTGCC; Pcsk9-1#(mouse): sense: CACCGCCGCCGGGGCTATGTCATCA; antisense: AAACTGATGACAT AGCCCCGGCGGC; Pcsk9-2#(mouse): sense: CACCGATGCTGGGA-TAATTCGCTCC; antisense: AAACGGAGCGAATTATCCCAGCATC.

**lentiviral-shRNA interference**
Puromycin-resistance shRNA plasmids were purchased from Suzhou GenePharma. shRNA plasmids were co-transfected with pMD2.G (#12259) and psPAX2 (#12260) into 293T cells to generate lentiviruses. After 2 days the virus-containing supernatant was collected and concentrated. Cells were infected with the lentiviruses supplemented with 4 µg/ml Polybrene. The puromycin selection (2.5 µg/mL) started 2 days after the infection and was continued for a week to generate stable cell pools. The sequences for each shRNA are as follows: sh-β-catenin(mouse): 5'-GCTGATATTGACGGGCAGTAT; sh-β-catenin(human): 5'-GCTTGGAATGAGACTGCTGAT.

**Fig. 7 | Pcsk9 is a direct target of Lin28b/let-7.** Before the experiment started, CAFs were cultured with 15376T-CM for 6 days to induce Lin28b expression. CM for CAFs culture was replaced daily (**d–j**, **m–p**). **a–d** 14837CAFs (**a**, **b**), Lin28b-WT-expressing 14837CAFs (**a**, **b**), 15376CAFs (**c**, **d**), and Lin28b-WT-expressing 15376CAFs (**c**, **d**) were cultured under high glucose (25 mM) or low glucose (2 mM). The levels of Lin28b and Pcsk9 were measured by western blotting (**a**, **c**). The levels of Pcsk9 in the supernatants were examined by ELISA (**b**, **d**). Representative of $n = 3$ independent experiments (**a**, **c**). **e**, **f** 15376CAFs and Fzd4-KO 15376CAFs were cultured under high glucose (25 mM) or low glucose (2 mM). The levels of Fzd4, Lin28b, and Pcsk9 were measured by western blotting (**e**). The levels of Pcsk9 in the supernatants were examined by ELISA (**f**). Representative of $n = 3$ independent experiments (**e**). **g**, **h** Fzd4$^+$ CAFs and Fzd4$^-$ CAFs were cultured under high glucose (25 mM) or low glucose (2 mM). the levels of Fzd4, Lin28b, and Pcsk9 were measured by western blotting (**g**). The levels of Pcsk9 in the supernatants were examined by ELISA (**h**). Representative of $n = 3$ independent experiments (**g**). **i** The levels of Pcsk9 in 15376CAFs, Lin28b-KO 15376CAFs, flag-Lin28b-WT(r)-expressing 15376CAFs and flag-Lin28b-MU(r)-expressing 15376CAFs were measured by western blotting. Representative of $n = 3$ independent experiments. **j** 15376CAFs were transfected with let-7a agomir or let-7 sponge vector. The protein levels of Lin28b and Pcsk9 were measured by western blotting. Representative of $n = 3$

independent experiments. **k**, **l** Sequence alignment of the putative let-7a binding sites, and sketch of the construction of wild-type or mutant Pcsk9 3′UTR (**k**). The relative luciferase activity was analyzed after the pmir-GLO-Pcsk9 3′UTR (wild-type or mutant) vectors were co-transfected into 293T cells with let-7a agomir or agomir nc (**l**). *P*-value by one-way ANOVA with Tukey's multiple comparison test. **m**, **n** The mRNA expression level (**m**) and mRNA stability (**n**) of Pcsk9 in 15376CAFs, Lin28b-KO 15376CAFs, flag-Lin28b-WT(r)-expressing 15376CAFs and flag-Lin28b-MU(r)-expressing 15376CAFs were measured by real-time qPCR. **o**, **p** 15376CAFs (**o**), Lin28b-KO 15376CAFs (**o**), flag-Lin28b-WT(r)-expressing 15376CAFs (**p**), and flag-Lin28b-MU(r)-expressing 15376CAFs (**p**) were collected and polysomes were fractionated on sucrose density gradients. Amount of Pcsk9 mRNA in various polysome fractions was analyzed by RT-PCR and normalized to 5S rRNA level. **q**, **r** 15376T cells were orthotopically injected into WT and *FSP-Cre;Lin28b$^{fl/fl}$* mice. After 2 weeks, the tumors were analyzed by Pcsk9 and α-SMA IHC staining. Representative images are shown. Scale bar: 30 μM (**q**). Pcsk9 IHC scores in CAFs were plotted ($n = 10$ views per group). Data are shown as mean±s.d. **r** Three biologically independent experiments were performed (**b**, **d**, **f**, **h**, **l–p**). Data are shown as mean ± s.d. *P*-value were determined by one-way ANOVA with Tukey's multiple comparison test (**b**, **d**, **f**, **h**, **l–p**) or two-tailed unpaired Student's t-tests (**r**).

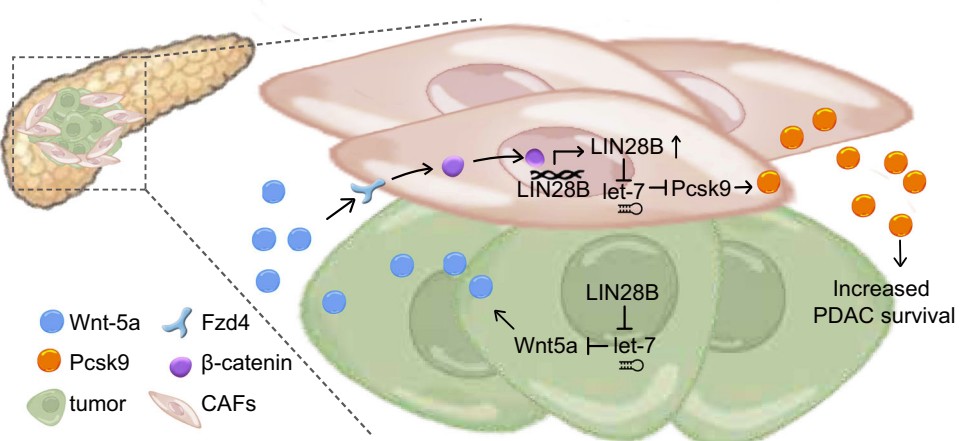

**Fig. 8 | Positive feedback loop between Wnt5a and Lin28b in PDAC.** Model of positive feedback loop between Wnt5a and Lin28b in PDAC. Tumor-secreted Wnt5a activates Wnt–β-catenin signaling pathway, inducing Lin28b expression in CAFs. Up-regulation of Lin28b in CAFs increases PDAC survival though promoting pcsk9 secretion.

## Overexpression plasmids

cDNA of Lin28b and Ldlr were amplified and cloned into p3×Flag-CMV-10. After that, the cDNA of flag-Lin28b and flag-Ldlr were cloned into lenti-EF1α-MCS-T2A-Hygro viral expression vector. The CRISPR/Cas9-resistant plasmids are synonymous mutations of a clustered randomly regularly interspaced short palindromic repeats (CRISPR)–Cas9 sequence with a Fast mutagenesis kit (Vazyme, #C214-01). the CRISPR/Cas9-resistant wild-type Lin28b (flag-Lin28b-WT(r)) plasmid sequence changes 'ATACGGGTAACAGGCCCAGG' to 'ATTCGCGTTACAGGCCCAGG'. The mutant Lin28b (flag-Lin28b-MU(r)) sequence are mutations of (flag-Lin28b-WT(r) sequence with a Fast mutagenesis kit (Vazyme, #C215-01). Five point-mutations (W34A, F43A, F61A, H159A and H181A) spread across the cold shock domain (CSD) and CysCysHisCys (CCHC) zinc-finger RNA-binding motifs were created in flag-Lin28b-MU(r) that render the protein incapable of let-7 suppression[21].

## Quantitative secretomics sample preparation and LC-MS/MS analysis

CAFs were cultured with 15376T-CM for 6 days and then they were counted and equal numbers of cells were split to 15 cm dishes at roughly 50% confluence. Once cell lines reached 70% confluence, two 15 cm dishes of each cell line were washed 3 times with PBS to remove

serum proteins and incubated for 24 h in 15 ml of FBS-free 15376T-CM. Conditioned media was then removed, mixed, filtered with 0.45 μm filters, and maximally concentrated (75 to 100-fold) via room temperature centrifugation at 4000 × *g* in 3-kDa molecular weight cutoff concentrating columns (Millipore). Concentrated samples were stored at −80 °C until processing for mass spectrometry.

Samples were separated via 10% SDS-PAGE and subjected to in-gel digestion. Briefly, the band of interest was excised from gel and cut into ~1 mm$^3$ cubes and destained with 50% acetonitrile (ACN) in 50 mM $NH_4HCO_3$. Proteins were reduced with dithiothreitol (DTT) and subsequent alkylation was performed with iodoacetamide (IAM). Trypsin-containing buffer was added to cover the gel pieces and incubated at 37 °C overnight. After digestion, peptides were extracted and dried completely in a dry vacuum pump and then resuspended in HPLC-grade water for LC–MS/MS analyses.

Tryptic digests were analyzed on a LTQ Velos Pro mass spectrometer (Thermo Scientific) equipped with nanoflow reversed phase liquid chromatography (EASY-nLC II, Thermo Scientific). Tryptic digests were loaded on a home-made capillary column (75 μm × 150 mm) with a laser-pulled electrospray tip (Model P-2000, Sutter instruments). The capillary column was packed with 5 μm, 100 Å Magic C18AQ silica-based particles (Michrom BioResources Inc., Auburn, CA). The mobile phase was composed of solvent A (97% $H_2O$, 3%

acetonitrile, and 0.1% formic acid) and solvent B (100% acetonitrile and 0.1% formic acid). The LC gradient consisted of 5% B for 3 min, a linear increase to 60% B over 50 min, a rapid increase to 90% B over 2 min, and 90% B for 10 min. The gradient was then switched back to 100% solvent A for column equilibration, and the flow rate was kept at 300 nl/min. Elution fractions from the capillary column were electrosprayed directly on to the mass spectrometer for MS and MS/MS analysis with data-dependent acquisition mode. One full MS scan ($m/z$ 400–1200) was acquired and the 10 most abundant precursor ions were selected for fragmentation in the LTQ by CID at a normalized collision energy setting of 35%.

## Secretomic data analyses
For data analysis, raw LC-MS files were searched against protein database of Mouse (Taxon identifier: 10090) using Mascot Deamon software (version 2.6.0, Matrix science). Precursor mass tolerance of peptides was set as 20 ppm, and for fragment ions in MS/MS scans the mass tolerance was set as 0.8 Da. Cysteine carbamidomethylation (C, 57.0215 Da) was set as a fixed modification and oxidation of methionine (M, 15.9949) was set as a variable modification. Trypsin was set as a digestion enzyme with a maximum of two missed cleavages. Quantitative secretomics analysis of 15376CAFs-CM and Lin28b-KO 15376CAFs-CM were provided in Supplementary Data file 4.

## RNA-seq analyses
The total RNA content of 15376T, Lin28b-KO 15376T, and 14837T was extracted as described above, and the library was prepared by Jiayin Biomedical Technology. Reads were then aligned using STAR v2.7.3.a. To remove low-quality/duplicate reads and sort the aligned BAM files, Samtools (v1.13) was used. Cufflinks, Cuffquant, and Cuffnorm v2.2.181 were used to assemble, quantify, and normalize transcripts (FPKM) to assemble the count tables, respectively.

## Whole-exome sequencing (WES) analyses
DNA of 15376T and 14837T was extracted and quantified as described above. Then WES using commercial DNA sequencing services by Mingma Technologies. Exome capturing was performed using NimbleGen SeqCap EZ Human Exome Library v3.0. Capture genomic DNAs were sequenced using Illumina NovaSeq 6000.

## Prediction of target genes of miRNAs
Targets of miRNAs are predicted by an online target prediction tool, TargetScan 8.0 (http:www.targetscan.org/), which predicts the biological targets of miRNAs by searching for the presence of conserved 8mer and 7mer sites that match the seed region of each miRNA.

## Chromatin immunoprecipitation (ChIP)
$2 \times 10^7$ cells were cross-linked with 1% formaldehyde, resuspended in lysis buffer on ice for 15 min and fragmented by sonication. Soluble chromatin was diluted and subjected to immunoprecipitation with the indicated antibodies. Immune complexes were precipitated with protein A/G Sepharose beads, washed sequentially with low-salt buffer (20 mM Tris-HCl, pH 8.0, 2 mM EDTA, 150 mM NaCl, 0.1% SDS, 1% Triton X-100), high-salt buffer (20 mM Tris-HCl, pH 8.0, 2 mM EDTA, 500 mM NaCl, 0.1% SDS, 1% Triton X-100), LiCl buffer (10 Mm Tris-HCl, pH 8.0, 1 mM EDTA, 250 mM LiCl, 1% Nonidet P-40, 1% sodium deoxycholate) and TE buffer (10 mM Tris-HCl, pH 8.0, 1 mM EDTA), and eluted with elution buffer (1% SDS and 0.1 M NaHCO₃). Cross-link was reversed and DNA was purified with DNA extraction kit (Vazyme). DNA was subjected to real-time PCR. Sequences for primers are: site 1: tcataaagaagcgtccc, atgctttcttgctacta; site 2: atctcttcctcttgcca, tcagcatcagtctaaca.

## Plasmid construction and luciferase assay
A 384 bp fragment of the Pcsk9 3'UTR (containing 2 predicted positions) was amplified by PCR using mouse genomic DNA as a template, and cloned into the pmir-GLO vector (Promega). Overlap PCR was used to construct 3'-UTR mutant reporter plasmid. The sequences of wild-type and mutant 3'-UTR were confirmed by sequencing. HEK 293T cells were seeded in 24-well plates and transfected with a mixture of pmir-GLO-Pcsk9 3'UTR (wild-type or mutant) and miRNA agomir (RiboBio, China) following the recommended protocol for the Lipofectamine 2000 transfection system (Invitrogen). Luciferase activity was assayed 2 days after transfection using the POLARstar Omega. TCF/LEF response element was cloned into the pGL6-TA vector (Beyotime Biotechnology). For transfecting 14837CAFs with TCF/LEF luciferase reporter plasmid and pRL-TK vector, neon transfection system (Invitrogen, MPK5000) was used following the manufacturer's instructions. Then, 14837CAFs were cultured with 14837T-CM or 15376T-CM in the presence or absence of 200 ng/ml r-mWnt5a. After 3 days, luciferase activity was assayed. Firefly luciferase activity was normalized to Renilla luciferase activity for each sample.

## mRNA stability analysis
Cells were treated with Actinomycin D to block RNA transcription, and RT-qPCR was used to assess the level of Pcsk9. After culturing at the various time points, cells were collected and RNA was extracted for RT-qPCR.

## Sucrose gradient fractionation of ribosomes
About $1 \times 10^7$ cells were washed twice in cold PBS and lysed by 1.0 ml polysome extraction buffer (PEB, 20 mM Tris-HCl [pH 7.5], 100 mM KCl, 5 mM MgCl₂, 0.5% NP-40) containing 100 μg/ml cycloheximide (CHX), 1 x protease inhibitors and 1:1000 dilution of RiboLock RNase inhibitor. The cell suspension was incubated on ice for 10 min, and centrifuged at $12,000 \times g$ for 15 min. Supernatant was collected and 300 μg of RNA was loaded onto the 10 to 50% sucrose gradients. Tubes were centrifuged using a Beckman SW40Ti rotor at $190,000 \times g$ for 1.5 h. Fractions were collected and mRNA was extracted with TRIzol. Analysis of mRNA distribution in sucrose gradient by qPCR.

## Immunohistochemical (IHC) staining
For IHC analysis, 3 mm sample sections were incubated with anti-Ki67 (1:1000), anti-CK19 (1:100), anti-α-SMA (1:100), and anti-Pcsk9 (1:100), respectively, overnight at 4 °C in a humidified chamber, followed by incubation with the HRP-conjugated secondary antibodies for 37 °C 60 min. Staining was completed by 1–2 min incubation with diaminobenzidine (DAB) substrate, which results in a brown-colored precipitate at the antigen site. The tissue sections stained immunohistochemically were analyzed and the mean staining intensity was calculated using ImageJ software.

## Immunofluorescence analysis
De-paraffined sections with 3-μm thickness were repaired by microwave. After that, the sections were treated with 0.5% Triton X-100 for 30 min and blocked with goat serum for 1 h at room temperature, washing with PBS 3× between each step. The sections were incubated with anti-α-SMA (1:100), anti-CK19 (1:100), anti-Lin28b (1:500), anti-Wnt5a (1:200), respectively, overnight at 4 °C in a humidified chamber. After incubation for 1 h with secondary antibodies, samples were incubated with DAPI. Anti-fade mounting medium (Yeasen, #36307ES25) was used to seal the sections. Immunofluorescent images were obtained using a confocal microscope.

## Masson's trichrome staining
Masson's trichrome staining was used to detect fibrosis. De-paraffined sections with 5-μm thickness were stained according to instructions (Applygen #B1130). Images were taken under uplight microscopy for analysis.

## Statistical analysis

Information about experimental replicates is included in the figure legends. Statistical analysis was performed using the SPSS statistical software package (standard v.20; SPSS Inc.). Human PDAC RNA-seq dataset from Clinical Proteomic Tumor Analysis Consortium (CPTAC) genomic data (dbGaP Study Accession: phs001287)[28] were used to test the correlation and gene expression signature score. The gene expression data and the clinical data were downloaded from GDC data portal (https://portal.gdc.cancer.gov/). The statistical analysis analyses were performed by R (Version: 4.1.3). Correlation between two continuous variables was measured by Pearson correlation coefficient. Different PDAC subtypes and iCAF/myCAF gene signature score analysis was performed using the R package "singscore" (https://bioconductor.org/packages/singscore/). Gene sets representing different PDAC subtypes markers and iCAF/myCAF markers were obtained from recent publications[26,27] (Supplementary data files 1 and 2). For two-group statistical analyses, unpaired Student's t-test was used. One-way analysis of variance followed by Tukey corrected post hoc t-test was used for multi-group comparisons. Error bars represent s.d., as indicated in the legends. Littermate animals from different cages were randomly assigned to the experimental groups. Tumor weight measurements were performed in a blinded manner. All other data collection and analysis were not performed blind to the conditions of the experiments. Data distribution was assumed to be normal but not formally tested. No data were excluded from the analyses.

No statistical method was used to predetermine sample size in all the highly controlled in vitro or in vivo experiments, but our sample sizes are similar to those reported in previous publications. For each experiment, we aimed for a number of at least of three samples or animals per group to allow basic statistical significance. The exact sample size was indicated in the figure legend. All animals were randomized and exposed to the same environment.

## Reporting summary

Further information on research design is available in the Nature Portfolio Reporting Summary linked to this article.

# Data availability

The RNA-Seq and whole-exome sequencing (WES) data generated in this study have been deposited in the NCBI's Gene Expression Omnibus database under accession code GSE217378. Secretomics data have been deposited to the ProteomeXchange Consortium via the iProX with the dataset identifier PXD045477. Source data used in the main figure and supplemental figure panels are also provided. Source data are provided with this paper.

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

## Acknowledgements

The authors thank Dr. Haoqiang Ying for kindly providing 14837T, 14838T, and 15376T. This study was supported by the National Natural Science Foundation of China (82173020, 81874145, and 81672712) to Y.Z., National Key R&D Program of China (2017YFA0503900, 2019YFC1005200) to Y.Z., Peking University (BMU2022XKQ004, PKU2020LCXQ024) to Y.Z.

## Author contributions

Z.S. carried out in vitro cell experiments, in vivo experiments, manuscript preparation, IHC analysis and statistical analysis; M.F., B.T., Haimeng.L., M.Y., J.B., S.H., Hengchao.L., and L.W. carried out in vitro cell experiments and in vivo experiments; H.L. provided clinical specimens; Z.T. and X.L. carried out secretome experiments and analyzed the data; W.W. carried out flow cytometry sorting experiments; H.W. and S.S. analyzed the data of whole-exome sequencing (WES); WG.Z. supervised the study. Y.Z. conceived, designed, supervised the study and wrote the manuscript. All authors read and approved the final version of the revised manuscript.

## Competing interests

The authors declare no competing interests.
