## [Peer Review File · Nature Communications]

The Lin28b/Wnt5a axis drives pancreas cancer through crosstalk between cancer associated fibroblasts and tumor epitheliumREVIEWER COMMENTS

Reviewer #1 (Remarks to the Author):

REVIEWER' UNDERSTANDING AND GENERAL COMMENTS

In this manuscript Shu et al. report on the role of Lin-28-B in favouring the establishment of a pro-tumorigenic cancer cells-stroma axis in pancreatic cancer (PC). A dense fibrotic reaction with abundant cancer-associated fibroblasts (CAFs) is a cornerstone of pancreatic cancer tissues. A multitude of CAFs phenotypes has been reported in PC. Some of the CAFs subtypes exert both pro- and anti-tumorigenic function (e.g., myCAFs), while others (e.g., iCAFs) have been consistently associated with pro-tumorigenic functions. Here, the authors focused on the role of tumoral Lin-28-B in driving the emergence of a pro-tumorigenic CAFs phenotype defined by Lin-28-B expression and sustaining tumour cell proliferation through the secretion of PCSK9, a protein with a well-defined enzymatic activity towards LDL receptors. The expression and the functional role of Lin-28-B have been already investigated in PC. In particular, the expression of Lin-28-B has been shown to promote PC aggressiveness through cell autonomous mechanisms. The interesting aspect of this work is that the authors have focused their attention on non-cell autonomous effect of Lin-28-B. First, they evaluated the protein expression of Lin-28-B in a cohort of 80 PC tissue specimens. They found that about 1/4 of the cases presented with high expression of Lin-28-B, which was not restricted to the tumour epithelium but observed also in the tumour stroma. In keeping with previous reports, the elevated tissue expression of Lin-28-B was associated to unfavourable outcomes. The finding of Lin-28-B expression in the stroma is interesting, but it could be better substantiated for example availing of sc-RNA-Seq data of human PC tissues as well as by performing co-localization immunofluorescence experiments. Based on the images provided, it appears to me that the immunohistochemical staining is quite noisy. Furthermore, staining of serial sections is not ideal to prove that a certain cell type co-express two or three different antigens (see for example Figure 3O).

Using in vitro models, the authors further showed that the direct co-cultivation with Lin28-Bhigh cancer cells or the exposure to their conditioned media induced the expression of Lin-28-B in CAFs. Since not all mouse primary PC cell lines expressed Lin-28-B, I am wondering whether the authors can confirm previous observations about Lin-28-B being induced by the loss of SIRT6. Lin28-Bhigh cells could also induce the expression of stromal Lin-28-B upon orthotopic transplantation in immunocompetent mice. Next, the authors moved to the identification of soluble mediator(s) that could be responsible for the induction of Lin-28-B in CAFs. They first looked at the genes differentially expressed between Lin-28-B proficient and deficient cancer cells. The DEG analysis evidenced, among others, two Wnt ligands (Wnt5a and Wnt10a). The authors mostly focused on demonstrating that Wnt5a can elicit Lin-28-B expression in CAFs in vitro and in vivo. However, it is not clear to me how Wnt-5A would induce Lin-28-B expression.

Then, the authors focused on demonstrating a pro-tumorigenic function of Lin-28-B expressing CAFs by evaluating their effect on cancer cell proliferation in vitro and in vivo. The in vivo experiment suggests that the Lin-28-B pro-tumorigenic effect in PC is mostly due to non-cell autonomous effect; Lin-28-B proficient and deficient cancer cells generate tumours similar of similar volumes in mouse defective for Lin-28-B in the stromal compartment. That finding is in stark contrast with previous reports, which have highlighted a cell autonomous effect of Lin-28-B in driving PC progression. I would suggest expanding the experiments with human preclinical models to better dissect whether the major contribution

of Lin-28-B to the progression of PC is via cell autonomous or non-cell autonomous mechanisms.

From the secretomic analysis of CAFs conditioned medium, the authors identified and functionally validated Pcsk9 as the soluble mediator, downstream of Lin-28-B in CAFs, which exerts pro-tumorigenic effect. The authors showed that Pcsk9 is a direct target of le7-a, which can reduce its level by blocking the translation rather than reducing Pcsk9 mRNA stability.

Overall, the findings of this work are of potential interest for the field as they expand the pro-tumorigenic role of Lin-28-B beyond what has already been reported. However, the negative aspects outweigh the positive ones in my opinion. The preparation of the manuscript is below average. There are a lot of missing (or incorrectly reported) information, which makes very difficult for the reader to navigate through the manuscript. In general, the conclusions are not always sufficiently supported by the evidence provided. I am also a bit sceptical about the experimental design, especially when it comes to the necessity of pre-exposing CAFs to conditioned medium from Lin-28-B high tumour cells before downstream analyses. Without a detail reporting about the procedures employed to mitigate the "carry-over" effect it is very difficult to judge on the quality of the results.

For example, how was the secretome analysis conducted. It is my understanding that the CAFs were pre-exposed to Lin-28-B high tumour cells to sustain the gene expression in the stromal cells. Indeed, the legends to the Figure 5 reads: "Before the experiment started, 15376 CAFs were co-cultured with 15376T at least 6 days to sustain Lin-28-B expression". First, it would be good to know whether it has been a direct coculture, a coculture using a tranwell assay or just the exposure to the conditioned medium. All these coculture systems are reported in the materials and method section, but which one has been used to perform that experiment is not clear. Second, in the section relative to the quantitative secretomics there is no mention of the coculture. Therefore, it is not clear whether the medium from the coculture or the monoculture has been assayed. And even if cells were first cocultured with cancer cells and then plated as monoculture, I would have expected that authors controlled for the expression changes in Lin-28-B following that procedure.

I also think that the authors should expand on the functional and potentially therapeutical relevance of Pcsk9. There are Pcsk9 inhibiting compound available which could be used to test whether Lin-28-B high tumours are really dependent on it.

One of my major concerns relates to the role of the tumour-derived Wnt5a as the soluble mediator that induces Lin-28-B in CAFs. The authors decided to focus on Wnt5a (why not Wnt10a) based on an RNA-seq experiment that shows increase abundance of Wnt5a mRNA in tumour cells expressing Lin-28-B as opposed to the KO as well as to an unrelated cell line that does not express Lin28b. If I am not mistaken, the RNA-Seq analysis has been conducted on 3 samples in total (one per genotype). If that is the case, how can we be sure about the repeatability of the results? Second, there is no conclusive mechanism about how Lin-28-B would regulate Wnt5a. The authors have availability of plasmids encoding for the wildtype and mutants Lin-28-B, which could be used to conclusively show that Lin-28-B influences the abundance of Wnt5a. Along the same line, it is not clear how Wnt5a increases the expression of Lin-28-B in CAFs. The authors have mostly relied their conclusion on coculture experiments (not sure it is coculture followed by FACS separation or exposure of CAFs to conditioned medium) and they showed that 6 days are needed to be able to elicit Lin-28-B expression. The kinetic of expression changes is suggestive of a Lin-28-B increase secondary to a wider reprogramming of cells rather than to a direct consequence of a ligand-to-receptor binding. The same kinetic is observed when experiments were conducted in the

presence of recombinant Wnt5a protein. In general, binding of wnt ligands to their receptor induces transient wnt-pathway activation. Therefore, I would be surprised if the LIN-28-B increased expression is the sole consequence of the Wnt5a binding to its receptors. Another point that should be addressed is why focusing on canonical wnt pathway activation. To the best of my knowledge, Wnt5a mostly, although not exclusively, acts through beta-catenin independent pathways. Beta-catenin is not the receptor for Wnt-5a (as erroneously reported in the text), and its knockdown will have phenotypic consequences, which are not necessarily related to the wnt pathway. Those have been overlooked here. What happened to the CAFs following inactivation of beta-catenin? Have you considered the genetic manipulation of other receptors (e.g., Fzd, Ror).

Something very confusing to me is the downregulation of PSCK9 in tumours resulting from the transplantation of LIN-28-Bhigh cancer cells in hosts lacking Lin-28-B in the stromal compartment or when those cells are co-transplanted with Psck9 KO CAFs. As shown by the authors, tumour tissues from LIN-28-Bhigh cancer cells showed intense Psck9 staining in both the tumour and the stromal compartment. That makes sense based on the data provided here: Psck9 as the target of let7a. While the expression of Lin-28-B is induced in CAFs, the mouse PDAC cell lines selected for experiments showed "constitutive" expression of the proteins. If Lin-28-B controls Psck9 expression, then tumour cells should express it. Other comments are provided below listed according to the figures of the manuscript

SPECIFIC COMMENTS

FIGURE 1.

I would suggest experienced pathologist(s) to review histological images. I am not quite sure that there are neoplastic cells in the images used for Lin28b-low tumours.

I think the quality of IHC staining is suboptimal and it is not entirely clear what one should expect to see. Where is Lin28b localized? Is the localization of Lin28b predominantly cytoplasmatic, exclusively cytoplasmatic? I can see nuclear staining. I would suggest using scRNA-Seq data to confirm about the presence of Lin28b in the stroma of PC patients. From the IHC images, it is very difficult to say whether CAFs is the only TME cell type with expression of Lin28b.

Figure 1C: I would include the actual number of patients in each category so that I can understand for example how many stage IV tumors are included.

Figure 1E, are we sure that is cancer?

FIGURE 2.

More data about the generation and characterization of CAFs should be provided. The generation of CAFs from mouse PC tumors can be difficult and data about their characterization needs to be provided (presence/absence of the engineered alleles) even if the cells were isolated from an iKRAS system and therefore dependent on doxycycline. Related to that, to explain separation of the two cell populations (i.e., CAFs and epithelial cells) by FACS, the authors state that primary PC cells are GFP positive while CAFs from wildtype mouse are not. First, there is no mention of the GFP allele when the mouse model is described. Second CAFs were not prepared from wildtype animals I hope, otherwise they are not CAFs. As discussed above, it is very difficult to understand whether the stromal and cancer cells have been co-cultured using a transwell system, a direct co-culture or it is just growing cells in conditioned medium. That is quite important to understand how the experiments were conducted downstream of the coculture.

E-H, have you have accounted for an effect of cell confluency on the expression of Lin28b? Is the increased Lin28b expression secondary to confluency, i.e. proliferation

FIGURE 3.

A major point here is to understand the numerosity of the samples subjected to RNA-seq, i.e. how many replicas per genotype. As also suggested above, the perturbation experiments with human cell lines should be expanded beyond the use of PANC1. Even if that is a widely used cell line, it is now well established that PANC1 is poorly representative of PC cells. Also, human cancers are very heterogeneous, and the generalization of results require the use of a wider panel of cell cultures.

Something that I think has been overlooked is whether the conditioned media from different tumour cell lines (with different levels of Lin-28-B) have different effect on the proliferation of CAFs and their phenotypic activation (myCAFvs vs iCAFvs)

FIGURE 4.

One general consideration: to avoid the problem of the pre-exposure of CAFs to cancer cells, I would have induced expression (e.g., ORF) of Lin-28-B in CAFs. That way, expression is not dependent on the epithelium.

Panel A, rescue experiments: in 25 mM glucose CAFs increases proliferation of tumor cells as opposed to the monoculture; not sure that the difference is significant as the statistical analysis is missing. In 2 mM of glucose, there is a significant drop in cancer cell proliferation which is expected. My question is whether the extent of proliferative reduction is the same in monoculture and coculture given that the coculture. Another point related to this experiment. Have the authors considered performing the rescue experiment with CAFs not expressing Lin-28-B? That would be important understand whether only Lin-28-B CAFs have the capability of sustaining cancer cells in harsh microenvironment.

Panel G. I urge the authors to do co-localization experiment as it is very difficult to understand whether aSMA + cells express or not Lin-28-B in the different conditions.

FIGURE 5.

Not sure why the focus became Pcsk9. It cannot just be that the levels of Psck9 are higher in CAFs cm than in the human cm. That is true also for other candidates.

There is a significant reduction of Psck9 staining in neoplastic cells (Figure 50). Why is that? Have you checked for the expression of Lin-28-B in these experiments? The lack of Lin-28-B in the tumour cells would also explain why there is no recruitment of resident fibroblasts that should indeed be Lin-28-B positive.

Reviewer #2 (Remarks to the Author):

In this manuscript, the authors describe a system of tumour-stroma crosstalk in pancreatic ductal adenocarcinoma. In particular, they argue that the RNA binding protein LIN28B is expressed in a subset of pancreatic tumours. LIN28B is concurrently expressed in tumour and stromal CAFs. LIN28B is a regulator of Wnt5a signal that, when secreted by the tumour cells, has an effect on neighbouring CAFs where LIN28B expression on its turn stimulates protein production of a certain cytokine, Pcsk9. Pcsk9 signals back to the tumour cells, supporting their proliferation. The study uses elegant in vitro and in vivo approaches but needs further refinement to support some conclusions and gain more detailed insights. The scope of the study should be broader (taking into account CAF heterogeneity, extrapolation to human samples, and inclusion of other elements from the microenvironment if this is

suggested by the title).

-The study is limited to “cancer associated fibroblasts”.

1) CAF heterogeneity in pancreatic cancer is widely described so the authors should investigate in detail which CAF population is under study. The fact that they use aSma as a CAF marker, and recent evidence shows that aSma+ myofibroblasts (one subtype of the CAFs) restrain the tumor rather than support its growth, is hard to reconcile with the data shown here.

2) The title does suggest a broader scope “tumor microenvironment”. If the authors want to keep such title, additional study of at least the immune microenvironment is necessary.

-The study is mostly limited to mouse work. At some point there is mentioning of the squamous subtype of human pancreatic cancer. Adding more detail on how all these genes/proteins are expressed in human tumors (and their subtypes) would greatly advance the study.

-Several analyses need to be refined.

1) Markers need to be assessed by double labelling to ascertain the cell type that is studied (e.g. for pKi67 – one cannot just assume that it are tumour cells that proliferate)

2) How are tumours exactly weighed after orthotopic grafting of the tumour cells?

3) KO of genes need to be validated.

4) What is the receptor for Pcsk9 and is it expressed in the tumour cells? This is essential to show.

5) What would have caused the divergence of 3 tumour cell lines from one genetically identical mouse line? Investigating this will add an extra level of information on tumor heterogeneity.

- Beta-catenin is mentioned as a Wnt receptor which is not correct.

- The discussion mentions elements that were not obvious in the results section, such as the presumed role of cholesterol.

Reviewer #3 (Remarks to the Author):

In this research manuscript, the authors present evidence that there is a feedback loop where LIN28B expression in pancreatic ductal adenocarcinoma (PDAC) induces WNT5A expression that in turn induces LIN28B expression in cancer associated fibroblasts (CAFs) that in turn secretes the cytokine PCSK9 that acts on the cancer cells.

Overall, this research is quite solid with appropriate controls and well-justified conclusions that uses multiple methods to reach their conclusions.

Overall summary:

Initially, they perform immunohistochemistry on a 90 patient cohort that revealed that LIN28B expression in cancers cells and stroma are tightly correlated where LIN28B high expression is correlated with poor prognosis. This association is seen in multiple cancer types where LIN28A/LIN28B expression is almost invariably correlated with poor survival. However, the connection to the stroma and ultimately CAFs is novel.

Next, they establish that a secreted factor from LIN28B-expressing cancer cells induces LIN28B expression in CAFs using biochemical and cell line approaches where the

secretome analysis using CRISPR modified cell lines that the soluble factor was WNT5A where recombinant WNT5A can induce LIN28B expression in CAFs.

Further work showed that LIN28B expression in CAFs upregulated a number of genes. This result is consistent with the widely published result that LIN28B negatively regulates biogenesis of the tumor suppressor let-7 microRNA family. The cytokine PCSK9 was selected for further study as this gene is predicted to be a let-7 regulated gene. The authors show both overexpression and inhibition experiments showing that PCSK9 is regulated by let-7. Further work shows that LIN28/let-7 likely regulates PCSK9 at the level of translational inhibition rather than mRNA degradation.

Minor points:

1. Figure 5E, the y-axis is miRNA I assume. Also, it is not clear what this data is normalized to.
2. For Figure 5 and the text, the authors most likely need to rename things. Initially, I thought LIN28-WT and LIN28B-MU lines that had both endogenous LIN28B expression and overexpression of the indicated protein. It turns out that this was a LIN28B CRISPR/Cas9 knockout line where an add back experiment was done. This is confusing.
3. Please update the manuscript using the approved gene nomenclature (e.g., human LIN28B, WNT5A,... and mouse Lin28b, Wnt5a, etc...)
<https://www.genenames.org/>
and
<http://informatics.jax.org/mgihome/nomen/>

Reviewer #1 (Remarks to the Author):

REVIEWER' UNDERSTANDING AND GENERAL COMMENTS

In this manuscript Shu et al. report on the role of Lin-28-B in favouring the establishment of a pro-tumorigenic cancer cells-stroma axis in pancreatic cancer (PC). A dense fibrotic reaction with abundant cancer-associated fibroblasts (CAFs) is a cornerstone of pancreatic cancer tissues. A multitude of CAFs phenotypes has been reported in PC. Some of the CAFs subtypes exert both pro- and anti-tumorigenic function (e.g., myCAFs), while others (e.g., iCAFs) have been consistently associated with pro-tumorigenic functions. Here, the authors focused on the role of tumoral Lin-28-B in driving the emergence of a pro-tumorigenic CAFs phenotype defined by Lin-28-B expression and sustaining tumour cell proliferation through the secretion of PCSK9, a protein with a well-defined enzymatic activity towards LDL receptors. The expression and the functional role of Lin-28-B have been already investigated in PC. In particular, the expression of Lin-28-B has been shown to promote PC aggressiveness through cell autonomous mechanisms. The interesting aspect of this work is that the authors have focused their attention on non-cell autonomous effect of Lin-28-B. First, they evaluated the protein expression of Lin-28-B in a cohort of 80 PC tissue specimens. They found that about 1/4 of the cases presented with high expression of Lin-28-B, which was not restricted to the tumour epithelium but observed also in the tumour stroma. In keeping with previous reports, the elevated tissue expression of Lin-28-B was associated to unfavourable outcomes. The finding of Lin-28-B expression in the stroma is interesting, but it could be better substantiated for example availing of sc-RNA-Seq data of human PC tissues as well as by performing co-localization immunofluorescence experiments. Based on the images provided, it appears to me that the immunohistochemical staining is quite noisy. Furthermore, staining of serial sections is not ideal to prove that a certain cell type co-express two or three different antigens (see for example Figure 3O).

Answer: Yes, we have changed the IHC staining to immunofluorescence staining in Fig.1e, Fig.1h, Fig.2s, Fig.3t and Fig.5q

We tried different single-cell sequencing database based PDAC patients, however, the limited depth and coverage of single-cell sequencing technique makes it is difficult to detect the expression level of LIN28B in any cell lineages of PDAC. Also, see below, as David Ting mentioned in his Nature Communication paper 2020, "Consistent with previous CTC enriched markers, LIN28B cancer cells comprise a small subpopulation of primary tumor cells, which suggests the acquisition of LIN28B transcriptional programs are unique to a select number of cells with metastatic propensity." Moreover, as shown below, LIN28B has inter- and intra-tumoral heterogeneity.

Supplementary Figure 7: LIN28B expression in resected PDAC tissue is heterogeneous. (a) RNA in situ hybridization (RNA-ISH) staining of resected PDAC tissue demonstrates “positive” (top micrograph) and “negative” (bottom micrograph) *LIN28B* staining within glandular PDAC cells. (b) Immunohistochemistry staining for LIN28B in resected PDAC tissue also demonstrates protein heterogeneity even between different glands within a given specimen. These results were confirmed on at least 5 resection specimens, confirming a range of RNA (a) and protein (b) expression.

Supplementary Figure 5: Immunofluorescent staining of CTCs from an independent cohort of patients with metastatic PDAC demonstrates heterogeneous nuclear expression of LIN28B. (a) Indirect immunofluorescence images of CTCs purified with the CTC-iChip, fixed, spun onto slides, and imaged. Pan-cytokeratin (CK) is pseudocolored in green, LIN28B in yellow, DNA from DAPI staining in blue, and CD45 in red. A CD45+ leukocyte derived from one of the patients is shown in the first “row” as a positive control for CD45 staining. This experiment was not repeated. (b) Single-cell data from prior work⁹, showing 17 micro-manipulated human PDAC CTCs showing expression of *KRT*, *MUC*, leukocyte (*PTPRC*, *CSF3R*, *CD14*, *ITGA2B*, *ITGB3*), and stem cell (*LIN28B*, *KLF4*, *WNT5A*, *LGALS3*) genes. Data are presented in RPM, with red

Franses JW, .., Ting DT. Pancreatic circulating tumor cell profiling identifies LIN28B as a metastasis driver and drug target. *Nat Commun.* 2020 Jul 3;11(1):3303.

Using in vitro models, the authors further showed that the direct co-cultivation with Lin28-Bhigh cancer cells or the exposure to their conditioned media induced the expression of Lin-28-B in CAFs. Since not all mouse primary PC cell lines expressed Lin-28-B, I am wondering whether the authors can confirm previous observations about Lin-28-B being induced by the loss of SIRT6. Lin28-Bhigh cells could also induce the expression of stromal Lin-28-B upon orthotopic transplantation in immunocompetent mice. Next, the authors moved to the identification of soluble mediator(s) that could be responsible for the induction of Lin-28-B in CAFs. They first looked at the genes differentially expressed between Lin-28-B proficient and deficient cancer cells. The DEG analysis evidenced, among others, two Wnt ligands (Wnt5a and Wnt10a). The

authors mostly focused on demonstrating that Wnt5a can elicit Lin-28-B expression in CAFs in vitro and in vivo. However, it is not clear to me how Wnt-5A would induce Lin-28-B expression.

Answer: As the reviewer suggested, we observed whether Lin28b could be induced by the loss of SIRT6. As shown in Supplementary Fig.7a-d, SIRT6 expression level is negatively correlated with Lin28b in both human PDAC and murine PDAC. However, knocking-out SIRT6 did not induce Lin28b expression in 14837T cells even in the presence of 15376T conditioned medium or recombinant-Wnt5a treatment (Supplementary Fig.7e). The first reason probably because of the difference of mouse models people used. Our 14837T cells were isolated from B6 background PDAC model (provided by Haoqiang Ying, Cell, 2012). It is a transgenic Kras mutation PDAC model, however, in “SIRT6 Suppresses Pancreatic Cancer through Control of Lin28b” paper, they used conventional KPC model which is a Kras G12D point mutation PDAC model. As David Tuveson mentioned in his paper (DeNicola GM et al, Nature. 2011), physiological expression of oncogenes lowers ROS level in MEF, but overexpression of mutant Kras boosts the level of ROS. Secondly, as David Ting discussed (Franses JW et al., Nat Commun.2020, see below), possibly due to multiple alternative pathways of lin28b regulation, the compensation mechanism makes the correlation between Lin28b and SIRT6 unclear.

supplementary Figure 6: LIN28B expression in PDAC CTCs correlates with several known associated driver genes. (a) LIN28B mRNA expression correlates with PDX1, a known pancreatic transcription factor. (b) SIRT6 and cMYC do not correlate with LIN28B expression in PDAC CTCs. (c) miR100HG, a lncRNA driven by TGF β signaling and associated with LIN28B, but not TGF β 1 or TGF β R1, correlates with LIN28B expression in PDAC CTCs. (d) LIN28B

Franses JW, ..., Ting DT. Pancreatic circulating tumor cell profiling identifies LIN28B as a metastasis driver and drug target. Nat Commun. 2020 Jul 3;11(1):3303.

Although knocking-out SIRT6 did not induce Lin28b expression in 14837T cells, we found that SIRT6 loss in CAFs could induce Lin28b expression (Supplementary Fig.7f). Since our revision data indicated that Wnt5a-Fzd4 pathway is essential for activation of Lin28b (Fig.5c-f), we then tested the protein level of Fzd4 in 14837T and 14837CAFs and found Fzd4 which could be expressed in 14837CAFs but not 14837T (Supplementary Fig.7e-f). Knocking-out SIRT6 could induce Lin28b expression in 14837CAFs in the presence of 15376T conditioned medium or recombinant-Wnt5a treatment (Supplementary Fig.7f). Moreover, consistent with previous study (Liu et al, 2021), we found that Fzd4 expression was also induced by loss of SIRT6 (Supplementary Fig.7f), suggesting that Fzd4 mediated SIRT6 -induced Lin28b expression.

DeNicola GM, Karreth FA, Humpton TJ, Gopinathan A, Wei C, Frese K, Mangal D, Yu KH, Yeo CJ, Calhoun ES, Scrimieri F, Winter JM, Hruban RH, Iacobuzio-Donahue C, Kern SE, Blair IA, Tuveson DA. Oncogene-induced Nrf2 transcription promotes ROS detoxification and tumorigenesis. *Nature*. 2011 Jul 6;475(7354):106-9.

Liu T, Li Z, Tian F. Quercetin inhibited the proliferation and invasion of hepatoblastoma cells through facilitating SIRT6-mediated FZD4 silence. *Hum Exp Toxicol*. 2021 Dec;40(12_suppl):S96-S107.

Then, the authors focused on demonstrating a pro-tumorigenic function of Lin-28-B expressing CAFs by evaluating their effect on cancer cell proliferation in vitro and in vivo. The in vivo experiment suggests that the Lin-28-B pro-tumorigenic effect in PC is mostly due to non-cell autonomous effect; Lin-28-B proficient and deficient cancer cells generate tumours similar of similar volumes in mouse defective for Lin-28-B in the stromal compartment. That finding is in stark contrast with previous reports, which have highlighted a cell autonomous effect of Lin-28-B in driving PC progression. I would suggest expanding the experiments with human preclinical models to better dissect whether the major contribution of Lin-28-B to the progression of PC is via cell autonomous or non-cell autonomous mechanisms.

Answer: Yes, thanks for the reviewers' comments, we have used several human cell lines and found PANC1 do have dependency on LIN28B's function as reported before (see below).

From the secretomic analysis of CAFs conditioned medium, the authors identified and functionally validated Pcsk9 as the soluble mediator, downstream of Lin-28-B in CAFs, which exerts pro-tumorigenic effect. The authors showed that Pcsk9 is a direct target of Ie7-a, which can reduce its level by blocking the translation rather than reducing Pcsk9 mRNA stability.

Overall, the findings of this work are of potential interest for the field as they expand the pro-tumorigenic role of Lin-28-B beyond what has already been reported. However, the negative aspects outweigh the positive ones in my opinion. The preparation of the manuscript is below average. There are a lot of missing (or incorrectly reported) information, which makes very difficult for the reader to navigate through the manuscript. In general, the conclusions are not always sufficiently supported by the evidence provided. I am also a bit sceptical about the experimental design, especially when it comes to the necessity of pre-exposing CAFs to conditioned medium from Lin-28-B high tumour cells before downstream analyses. Without a detail reporting about the procedures employed to mitigate the "carry-over" effect it is very difficult to judge on the quality of the results.

For example, how was the secretome analysis conducted. It is my understanding that the CAFs were pre-exposed to Lin-28-B high tumour cells to sustain the gene expression in the stromal cells. Indeed, the legends to the Figure 5 reads: "Before the experiment started, 15376CAF were co-cultured with 15376T at least 6 days to sustain Lin-28-B expression". First, it would be good to know whether it has been a direct coculture, a coculture using a transwell assay or just the exposure to the conditioned medium. All these coculture systems are reported in the materials and method section, but which one has been used to perform that experiment is not clear. Second, in the section relative to the quantitative secretomics there is no mention of the coculture. Therefore, it is not clear whether the medium from the coculture or the monoculture has been assayed. And even if cells were first cocultured with cancer cells and then plated as monoculture, I would have expected that authors controlled for the expression changes in Lin-28-B following that procedure.

Answer: 15376wt-CAFs or Lin28b-KO-CAFs were cultured with 15376T CM for 6 days and then they were cultured with FBS-free DMEM for 1 day. We next performed quantitative secretomics analysis by using CM from 15376CAFs or Lin28b-KO 15376CAFs to identify the cytokines that were secreted from Lin28b-positive CAFs (Supplementary Table 4). As the reviewer suggested, we also tested the expression level of Lin28b and found that it was not affected by the culture condition (Supplementary Fig.4a).

I also think that the authors should expand on the functional and potentially therapeutical relevance of Psck9. There are Psck9 inhibiting compound available which could be used to test whether Lin-28-B high tumours are really dependent on it.

Answer: Thanks for the constructive suggestions. We examined the efficacy of PCSK9 inhibitor alirocumab in an orthotopic model of Lin28b^{high} 15376T and found that alirocumab resulted in significant tumor growth inhibition (Fig.6u).

One of my major concerns relates to the role of the tumour-derived Wnt5a as the soluble mediator that induces Lin-28-B in CAFs. The authors decided to focus on Wnt5a (why not Wnt10a) based on an RNA-seq experiment that shows increase abundance of Wnt5a mRNA in tumour cells expressing Lin-28-B as opposed to the KO as well as to an unrelated cell line that does not express Lin28b. If I am not mistaken, the RNA-Seq analysis has been conducted on 3 samples in total (one per genotype). If that is the case, how can we be sure about the repeatability of the results?

Answer: In our study, knocking-out Wnt5a but not Wnt10a in tumors suppressed the ability of tumor cells to induce Lin28b expression in CAFs, therefore we confirmed the role of Wnt5a in driving Lin28b expression in CAFs (Fig.3h-i and Supplementary Fig.2i-j).

As the reviewer suggested, we added samples for RNA-seq (3 replicas per genotype). The RNA-seq data also showed that Wnt5a was upregulated in 15376T compared with Lin28b-KO 15376T and 14837T.

Second, there is no conclusive mechanism about how Lin-28-B would regulate Wnt5a. The authors have availability of plasmids encoding for the wildtype and mutants Lin-28-B, which could be used to conclusively show that Lin-28-B influences the abundance of Wnt5a.

Answer: To address this comment, we used wild-type/mutant Lin28b plasmids and found that Lin28b WT, but not Lin28b mutant, increased Wnt5a protein level which was suppressed by Lin28b knocking-out (Supplementary Fig.6a). The rescue experiment proved that Wnt5a is a reliable target of Lin28b. In addition, Wnt5a was increased by let-7 sponge, indicating that Wnt5a was able to be regulated by Lin28b/let-7 pathway (Supplementary Fig.6b). Moreover we observed that Lin28b/let-7 pathway is able to regulate Wnt5a mRNA stability in PDACs (Supplementary Fig.6c).

Along the same line, it is not clear how Wnt5a increases the expression of Lin-28-B in CAFs. The authors have mostly relied their conclusion on coculture experiments (not sure it is coculture followed by FACS separation or exposure of CAFs to conditioned medium) and they showed that 6 days are needed to be able to elicit Lin-28-B expression. The kinetic of expression changes is suggestive of a Lin-28-B increase secondary to a wider reprogramming of cells rather than to a direct consequence of a ligand-to-receptor binding. The same kinetic is observed when experiments were conducted in the presence of recombinant Wnt5a protein. In general, binding of wnt ligands to their receptor induces transient wnt-pathway activation. Therefore, I would be surprised if the LIN-28-B increased expression is the sole consequence of the Wnt5a binding to its receptors.

Answer: We collected conditioned medium from 15376T with different confluency (30%-80%). As shown in Fig.2e-j, CM from 15376T with 80% confluency could induce Lin28b expression in CAFs in 2 days, while the similar effect was observed in 5-6 days if CM from 15376T with 30% confluency was used. We hypothesized that the different kinetic is related with Wnt5a amount in the CM. To test it, we examined the Wnt5a level in 15376T CM and found that Wnt5a is increased by high confluency of cells (Fig.3m). The similar kinetic of Lin28b expression was also observed in CAFs treated with different concentration of recombinant-Wnt5a (Fig.3n-q). Also, we generated Fzd4-KO (Wnt5a receptor) CAF, and found that Fzd4 depletion abolished the Wnt5a dependent reprogramming (Fig.4c-d).

Another point that should be addressed is why focusing on canonical wnt pathway activation. To the best of my knowledge, Wnt5a mostly, although not exclusively, acts through beta-catenin independent pathways. Beta-catenin is not the receptor for Wnt-5a (as erroneously reported in the text), and its knockdown will have phenotypic consequences, which are not necessarily related to the wnt pathway. Those have been overlooked here. What happened to the CAFs following inactivation of beta-catenin?

Have you considered the genetic manipulation of other receptors (e.g., Fzd, Ror).

Answer: Wnt5a has traditionally been known as a non-canonical WNT. However, in addition to activation of the non-canonical signaling pathways, Wnt5a was reported to activate the canonical pathway in the presence of Fzd4 receptors (Mikels et al, 2006; Jiang et al, 2022). In our study, we found that knocking-out Fzd4 in CAFs attenuated Lin28b expression induced by r-Wnt5a or CM from Lin28b^{high} tumors (Fig.4c-d). Moreover, r-Wnt5a or 15376T CM can induce modest β -catenin protein accumulation (Fig.4g-i) and enrichment of β -catenin in Lin28b promoters (Fig.4j-l) in CAFs.

In addition, we have corrected “ β -catenin is the receptor for Wnt-5a” into “ β -catenin is an intracellular signal transducer in the Wnt pathway” in our manuscript.

Mikels AJ, Nusse R. Purified Wnt5a protein activates or inhibits beta-catenin-TCF signaling depending on receptor context. *PLoS Biol.* 2006 Apr;4(4):e115.

Jiang X, Liu J, Guan Y, Zhao Z, Meng F, Wang X, Gao X, Zhou F, Chen Y, Wang X. The mechanism of the WNT5A and FZD4 receptor mediated WNT/ β -catenin pathway in the degeneration of ALS spinal cord motor neurons. *Biochem Biophys Res Commun.* 2022 Jun 18;609:23-30

Something very confusing to me is the downregulation of PSCK9 in tumours resulting from the transplantation of LIN-28-Bhigh cancer cells in hosts lacking Lin-28-B in the stromal compartment or when those cells are co-transplanted with Psck9 KO CAFs. As shown by the authors, tumour tissues from LIN-28-Bhigh cancer cells showed intense Psck9 staining in both the tumour and the stromal compartment. That makes sense based on the data provided here: Psck9 as the target of let7a. While the expression of Lin-28-B is induced in CAFs, the mouse PDAC cell lines selected for experiments showed "constitutive" expression of the proteins. If Lin-28-B controls Psck9 expression, then tumour cells should express it.

Answer: Previous studies have showed that Psck9 can be internalized by cells (Lagace et al., 2006; Fisher et al., 2007). Psck9 secreted by CAFs can also be uptaken by PDAC cells. The reduced staining of Psck9 staining was observed in PDACs which were co-injected with Psck9-KO CAFs (Fig.6q), suggested that CAFs are the main source of Psck9 in this model.

Lagace TA, Curtis DE, Garuti R, McNutt MC, Park SW, Prather HB, Anderson NN, Ho YK, Hammer RE, Horton JD. Secreted PCSK9 decreases the number of LDL receptors in hepatocytes and in livers of parabiotic mice. *J Clin Invest.* 2006 Nov;116(11):2995-3005.

Fisher TS, Lo Surdo P, Pandit S, Mattu M, Santoro JC, Wisniewski D, Cummings RT, Calzetta A, Cubbon RM, Fischer PA, Tarachandani A, De Francesco R, Wright SD, Sparrow CP, Carfi A, Sitlani A. Effects of pH and low density lipoprotein (LDL) on PCSK9-dependent LDL receptor regulation. *J Biol Chem.* 2007 Jul 13;282(28):20502-12.

Other comments are provided below listed according to the figures of the manuscript

SPECIFIC COMMENTS

FIGURE 1.

I would suggest experienced pathologist(s) to review histological images. I am not quite sure that there are neoplastic cells in the images used for Lin28b-low tumours.

Answer: Thank you for the suggestion. We invited pathologists to reviewed the histological images and chose more typical images in Fig.1a.

I think the quality of IHC staining is suboptimal and it is not entirely clear what one should expect to see. Where is Lin28b localized? Is the localization of Lin28b predominantly cytoplasmatic, exclusively cytoplasmatic? I can see nuclear staining. I would suggest using scRNA-Seq data to confirm about the presence of Lin28b in the stroma of PC patients. From the IHC images, it is very difficult to say whether CAFs is the only TME cell type with expression of Lin28b.

Answer: LIN28B was found in both the cytoplasm and the nucleus (Piskounova et al., 2011;Tao et al., 2020). Previous study showed that LIN28B is predominantly present in the cytoplasm of G1 phase cells and in nucleus of S phase and G2 phase cells (Guo et al., 2006). In addition, immunofluorescence experiments were performed to confirm the presence of Lin28b in the stroma (Fig.1e and 1h).

Piskounova E, Polytarchou C, Thornton JE, LaPierre RJ, Pothoulakis C, Hagan JP, Iliopoulos D, Gregory RI. Lin28A and Lin28B inhibit let-7 microRNA biogenesis by distinct mechanisms. *Cell*. 2011 Nov 23;147(5):1066-79.

Tao T, Shi H, Mariani L, Abraham BJ, Durbin AD, Zimmerman MW, Powers JT, Missios P, Ross KN, Perez-Atayde AR, Bulyk ML, Young RA, Daley GQ, Look AT. LIN28B regulates transcription and potentiates MYCN-induced neuroblastoma through binding to ZNF143 at target gene promoters. *Proc Natl Acad Sci U S A*. 2020 Jul 14;117(28):16516-16526.

Guo Y, Chen Y, Ito H, Watanabe A, Ge X, Kodama T, Aburatani H. Identification and characterization of lin-28 homolog B (LIN28B) in human hepatocellular carcinoma. *Gene*. 2006 Dec 15;384:51-61

Figure 1C: I would include the actual number of patients in each category so that I can understand for example how many stage IV tumors are included.

Answer: Yes, we have included the actual number of patients in each category in Fig.1c.

Figure 1E, are we sure that is cancer?

Answer: We stained CK19 as PDAC marker in Fig.1e.

FIGURE 2.

More data about the generation and characterization of CAFs should be provided. The generation of CAFs from mouse PC tumors can be difficult and data about their characterization needs to be provided (presence/absence of the engineered alleles) even if the cells were isolated from an iKRAS system and therefore dependent on doxycycline.

Answer: The presence of engineered alleles of tumor and CAFs were shown in Supplementary Fig.1b. We checked the Kras and Cre alleles in CAFs and using 14837T as a control.

Related to that, to explain separation of the two cell populations (i.e., CAFs and epithelial cells) by FACS, the authors state that primary PC cells are GFP positive while CAFs from wildtype mouse are not. First, there is no mention of the GFP allele when the mouse model is described.

Answer: The iKRAS mice were crossed to ROSA26-LSL-rtTA-IRES-GFP (ROSA_rtTA) and p48-Cre mice to enable pancreas-specific and doxycycline (doxy)-inducible expression of Kras^{G12D} (Ying et al., 2012, Cell). Then primary PC cells are isolated from this mice model and therefore they are GFP positive. We have added this description in the “method” section.

Ying H, Kimmelman AC, Lyssiotis CA, Hua S, Chu GC, Fletcher-Sananikone E, Locasale JW, Son J, Zhang H, Coloff JL, Yan H, Wang W, Chen S, Viale A, Zheng H, Paik JH, Lim C, Guimaraes AR, Martin ES, Chang J, Hezel AF, Perry SR, Hu J, Gan B, Xiao Y, Asara JM, Weissleder R, Wang YA, Chin L, Cantley LC, DePinho RA. Oncogenic Kras maintains pancreatic tumors through regulation of anabolic glucose metabolism. Cell. 2012 Apr 27;149(3):656-70.

Second CAFs were not prepared from wildtype animals I hope, otherwise they are not CAFs. As discussed above, it is very difficult to understand whether the stromal and cancer cells have been co-cultured using a transwell system, a direct co-culture or it is just growing cells in conditioned medium. That is quite important to understand how the experiments were conducted downstream of the coculture.

Answer: The details of the co-culture experiments were added in the figure legends.

We isolated mouse PSCs from PDAC orthotopic model according to previous study (Sous et al., 2016). PSCs become activated toward a myofibroblast phenotype which shows α -SMA expression upon adherence in 2D tissue culture (Apte et al., 1998). In our system, we tested α -SMA expression in isolated mouse PSCs to confirm stellate cells have transformed into myofibroblast-like cells. Actually, there is also some other group isolated CAFs from orthotopic

model in wild-type mice (Kim et al., 2022). In addition, to further validate our conclusion, we also isolate CAFs from KPC mice using FACS (Elyada et al., 2019). Then we used Fzd4 antibody to segregate CAFs into Fzd4-positive and Fzd4-negative populations. Lin28b's expression could be induced by tumor CM and Wnt5a in Fzd4-positive CAFs but not in Fzd4-negative CAFs (Fig.4f), indicating that intact Wnt5a-Fzd4 pathway is essential for activation of Lin28b.

Sousa, C., Biancur, D., Wang, X. et al. Pancreatic stellate cells support tumour metabolism through autophagic alanine secretion. *Nature* 536, 479–483 (2016).

Apte MV, Haber PS, Applegate TL, Norton ID, McCaughan GW, Korsten MA, Pirola RC, Wilson JS. Periacinar stellate shaped cells in rat pancreas: identification, isolation, and culture. *Gut*. 1998 Jul;43(1):128-33.

Kim, D.K., Jeong, J., Lee, D.S. et al. PD-L1-directed PIGF/VEGF blockade synergizes with chemotherapy by targeting CD141+ cancer-associated fibroblasts in pancreatic cancer. *Nat Commun* 13, 6292 (2022).

Elyada E, Bolisetty M, Laise P, Flynn WF, Courtois ET, Burkhart RA, Teinor JA, Belleau P, Biffi G, Lucito MS, Sivajothi S, Armstrong TD, Engle DD, Yu KH, Hao Y, Wolfgang CL, Park Y, Preall J, Jaffee EM, Califano A, Robson P, Tuveson DA. Cross-Species Single-Cell Analysis of Pancreatic Ductal Adenocarcinoma Reveals Antigen-Presenting Cancer-Associated Fibroblasts. *Cancer Discov*. 2019 Aug;9(8):1102-1123.

E-H, have you have accounted for an effect of cell confluency on the expression of Lin28b? Is the increased Lin28b expression secondary to confluency, i.e. proliferation

Answer: We tested the expression of Lin28b in CAFs which were cultured in conditioned medium or fresh medium up to 6 days. As shown in Supplementary Fig.1c-d, only conditioned medium from tumor could induced Lin28b expression, indicating that cell confluency had no effect on the expression of Lin28b.

FIGURE 3.

A major point here is to understand the numerosity of the samples subjected to RNA-seq, i.e. how many replicas per genotype.

Answer: As the reviewer suggested, we added samples for RNA-seq (3 replicas per genotype).

As also suggested above, the perturbation experiments with human cell lines should be expanded beyond the use of PANC1. Even if that is a widely used cell line, it is now well established that PANC1 is poorly representative of PC cells. Also, human cancers are very heterogeneous, and the generalization of results require the use of a wider panel of cell cultures.

Answer: We measured LIN28B expression in human PDAC cell lines and found that LIN28B was highly expressed in PANC-1 and PANC03.27, is not expressed by ASPC-1, Mia Paca-2, PaTu8988T and PaTu8988S (Fig.2m-n). Therefore we repeated our experiments in PANC03.27 and obtained the similar data (Fig.3k). In addition, we isolated human PDAC cell lines from 2 patients and found that LIN28B was highly expressed in hPDAC1[#] but not in hPDAC2[#] (Fig.2m-n). Consistently with the data from mPDAC, LIN28B expression in human CAFs is induced by conditioned medium from hPDAC1[#] tumor cells in a Wnt5a-dependent manner (Fig.3l).

Something that I think has been overlooked is whether the conditioned media from different tumour cell lines (with different levels of Lin-28-B) have different effect on the proliferation of CAFs and their phenotypic activation (myCAF_s vs iCAF_s)

Answer: The proliferation of CAFs cannot be affected by the conditioned media from 14837T or 15376T (Supplementary Fig.8a). We also examined the feature genes of myCAF_s and iCAF_s (Elyada et al, 2019) using RT-PCR and found that CAFs were not obviously switched from one subtype to another by the CM from 14837T or 15376T (Supplementary Fig.8b-c and Supplementary Table 2).

Elyada E, Bolisetty M, Laise P, Flynn WF, Courtois ET, Burkhart RA, Teinor JA, Belleau P, Biffi G, Lucito MS, Sivajothi S, Armstrong TD, Engle DD, Yu KH, Hao Y, Wolfgang CL, Park Y, Preall J, Jaffee EM, Califano A, Robson P, Tuveson DA. Cross-Species Single-Cell Analysis of Pancreatic Ductal Adenocarcinoma Reveals Antigen-Presenting Cancer-Associated Fibroblasts. *Cancer Discov.* 2019 Aug;9(8):1102-1123.

FIGURE 4.

One general consideration: to avoid the problem of the pre-exposure of CAFs to cancer cells, I would have induced expression (e.g., ORF) of Lin-28-B in CAFs. That way, expression is not dependent on the epithelium.

Panel A, rescue experiments: in 25 mM glucose CAFs increases proliferation of tumor cells as

opposed to the monoculture; not sure that the difference is significant as the statistical analysis is missing. In 2 mM of glucose, there is a significant drop in cancer cell proliferation which is expected. My question is whether the extent of proliferative reduction is the same in monoculture and coculture given that the coculture.

Answer: As the reviewer suggested, the statistical analysis was added in Fig.5a. The extent of proliferative reduction of tumor cells is the same in monoculture and cultured in 15376-CM (shown below).

Another point related to this experiment. Have the authors considered performing the rescue experiment with CAFs not expressing Lin-28-B? That would be important to understand whether only Lin-28-B CAFs have the capability of sustaining cancer cells in harsh microenvironment.

Answer: Thanks for the constructive suggestions. We generated stable 14837CAF and 15376CAF cell lines expressing WT-Lin28b (Supplementary Fig.3b-c). We then found that CM from CAFs expressing Lin28b was able to rescue tumor growth under glucose-limiting conditions (Fig.5c-d). Moreover, consistent with the *in vitro* proliferation data, tumor growth was significantly increased when co-injected with WT-Lin28b-expressing CAFs, and this increase was significantly attenuated when PDAC cells (14837T) were co-injected with 14837CAFs or 15376CAFs (Fig.5i-j). Furthermore, Pcsk9 levels in Lin28b-expressing CAFs CM were much higher than that in CM of CAFs not expressing Lin28b (Fig.7a-d), indicating that Pcsk9 secreted by Lin28b-positive CAFs promotes PDAC growth.

In addition, since our revision data indicated that Wnt5a-Fzd4 pathway is essential for activation of Lin28b (Fig.5c-f), we also tested the role of Fzd4 in tumor growth. As shown in Fig.5k-p, Fzd4-positive CAFs was able to rescue tumor growth both *in vitro* and *in vivo*, indicating that Fzd4-mediated lin28b expression in CAFs promotes PDAC growth.

Panel G. I urge the authors to do co-localization experiment as it is very difficult to understand whether aSMA + cells express or not Lin-28-B in the different conditions.

Answer: As the reviewer suggested, co-localization immunofluorescence experiments were performed. As shown in Fig.5q, stromal Lin28b was dramatically downregulated in FSP-Cre;Lin28b^{fl/fl} mice compared to that in WT mice.

FIGURE 5.

Not sure why the focus became Pcsk9. It cannot just be that the levels of Pcsk9 are higher in CAFs cm than in the human cm. That is true also for other candidates.

There is a significant reduction of Pcsk9 staining in neoplastic cells (Figure 50). Why is that? Have you checked for the expression of Lin-28-B in these experiments? The lack of Lin-28-B in the tumour cells would also explain why there is no recruitment of resident fibroblasts that should indeed be Lin-28-B positive.

Answer: Previous studies have showed that Pcsk9 can be internalized by cells (Lagace et al., 2006; Fisher et al., 2007). Pcsk9 secreted by CAFs can also be uptaken by PDAC cells. The reduced staining of Pcsk9 staining was observed in PDACs which were co-injected with Pcsk9-KO CAFs (Fig.6q), suggested that CAFs are the main source of Pcsk9 in this model. We also checked the expression of Lin28b in mPDAC and found that it was not affected by recombinant- Pcsk9 (Supplementary Fig.9a).

Lagace TA, Curtis DE, Garuti R, McNutt MC, Park SW, Prather HB, Anderson NN, Ho YK, Hammer RE, Horton JD. Secreted PCSK9 decreases the number of LDL receptors in hepatocytes and in livers of parabiotic mice. *J Clin Invest.* 2006 Nov;116(11):2995-3005.

Fisher TS, Lo Surdo P, Pandit S, Mattu M, Santoro JC, Wisniewski D, Cummings RT, Calzetta A, Cubbon RM, Fischer PA, Tarachandani A, De Francesco R, Wright SD, Sparrow CP, Carfi A, Sitlani A. Effects of pH and low density lipoprotein (LDL) on PCSK9-dependent LDL receptor regulation. *J Biol Chem.* 2007 Jul 13;282(28):20502-12.

Reviewer #2 (Remarks to the Author):

In this manuscript, the authors describe a system of tumour-stroma crosstalk in pancreatic ductal adenocarcinoma. In particular, they argue that the RNA binding protein LIN28B is expressed in a subset of pancreatic tumours. LIN28B is concurrently expressed in tumour and stromal CAFs. LIN28B is a regulator of Wnt5a signal that, when secreted by the tumour cells, has an effect on neighbouring CAFs where LIN28B expression on its turn stimulates protein production of a certain cytokine, Pcsk9. Pcsk9 signals back to the tumour cells, supporting their proliferation. The study uses elegant in vitro and in vivo approaches but needs further refinement to support some conclusions and gain more detailed insights. The scope of the study should be broader (taking into account CAF heterogeneity, extrapolation to human samples, and inclusion of other elements from the microenvironment if this is suggested by the title).

-The study is limited to “cancer associated fibroblasts”.

1) CAF heterogeneity in pancreatic cancer is widely described so the authors should investigate in detail which CAF population is under study. The fact that they use aSma as a CAF marker, and recent evidence shows that aSMA⁺ myofibroblasts (one subtype of the CAFs) restrain the tumor rather than support its growth, is hard to reconcile with the data shown here.

Answer: The question is very constructive. In fact, Lin28b can be expressed in a variety of CAFs including aSMA⁺ myofibroblasts and FAP⁺ myofibroblasts which support tumor growth (Fig.1i-j). Although previous study showed that aSMA⁺ myofibroblasts restrain the tumor, but most of these myCAF are functional in inhibiting the metastasis of PDAC, as to tumor growth, it is still controversial. We cannot exclude the possibility that some subtype of aSMA⁺ myofibroblasts might have the ability to promote tumor growth. For example, sc-RNAseq showed that several clusters of LRRC15⁺ myofibroblasts which promote tumor growth also highly expressed aSMA (Krishnamurty et al, 2022). Also, previous studies indicate that some subpopulation of myofibroblasts promotes the development and metastasis of PDAC (Sangrador I et al, 2018; Sun X et al, 2022).

Wnt5a was reported to activate the canonical pathway in the presence of Fzd4 receptors (Mikels et al, 2006; Jiang et al, 2022). In our study, we found that knocking-out Fzd4 in CAFs attenuated Lin28b expression induced by r-Wnt5a or CM from Lin28b^{high} tumors (Fig.4c-d). To further investigate the CAF heterogeneity, we isolate CAFs from KPC mice using FACS (Elyada et al., 2019) and then used Fzd4 antibody to segregate CAFs into Fzd4-positive and Fzd4-negative populations. Lin28b's expression could be induced by tumor CM and Wnt5a in Fzd4-positive CAFs but not in Fzd4-negative CAFs (Fig.4e-f), indicating that intact Wnt5a-Fzd4 pathway is essential for activation of Lin28b. We then found that Fzd4-positive CAFs was able to rescue

tumor growth both *in vitro* and *in vivo* (Fig.5k-p). Furthermore, Pcsk9 levels in Fzd4-positive CAFs CM were much higher than that in CM of Fzd4-negative CAFs (Fig.7e-h), indicating that Pcsk9 secreted by Fzd4-positive CAFs promotes PDAC growth.

Sun X, He X, Zhang Y, Hosaka K, Andersson P, Wu J, Wu J, Jing X, Du Q, Hui X, Ding B, Guo Z, Hong A, Liu X, Wang Y, Ji Q, Beyaert R, Yang Y, Li Q, Cao Y. Inflammatory cell-derived CXCL3 promotes pancreatic cancer metastasis through a novel myofibroblast-hijacked cancer escape mechanism. *Gut*. 2022 Jan;71(1):129-147.

Sangrador I, Molero X, Campbell F, Franch-Expósito S, Rovira-Rigau M, Samper E, Domínguez-Fraile M, Fillat C, Castells A, Vaquero EC. Zeb1 in Stromal Myofibroblasts Promotes Kras-Driven Development of Pancreatic Cancer. *Cancer Res*. 2018 May 15;78(10):2624-2637.

Krishnamurthy AT, Shyer JA, Thai M, Gandham V, Buechler MB, Yang YA, Pradhan RN, Wang AW, Sanchez PL, Qu Y, Breart B, Chalouni C, Dunlap D, Ziai J, Elstrott J, Zacharias N, Mao W, Rowntree RK, Sadowsky J, Lewis GD, Pillow TH, Nabet BY, Banchereau R, Tam L, Caothien R, Bacarro N, Roose-Girma M, Modrusan Z, Mariathasan S, Müller S, Turley SJ. LRRC15+ myofibroblasts dictate the stromal setpoint to suppress tumour immunity. *Nature*. 2022 Nov;611(7934):148-154.

Mikels AJ, Nusse R. Purified Wnt5a protein activates or inhibits beta-catenin-TCF signaling depending on receptor context. *PLoS Biol*. 2006 Apr;4(4):e115.

Jiang X, Liu J, Guan Y, Zhao Z, Meng F, Wang X, Gao X, Zhou F, Chen Y, Wang X. The mechanism of the WNT5A and FZD4 receptor mediated WNT/ β -catenin pathway in the degeneration of ALS spinal cord motor neurons. *Biochem Biophys Res Commun*. 2022 Jun 18;609:23-30

Elyada E, Bolisetty M, Laise P, Flynn WF, Courtois ET, Burkhart RA, Teinor JA, Belleau P, Biffi G, Lucito MS, Sivajothi S, Armstrong TD, Engle DD, Yu KH, Hao Y, Wolfgang CL, Park Y, Preall J, Jaffee EM, Califano A, Robson P, Tuveson DA. Cross-Species Single-Cell Analysis of Pancreatic Ductal Adenocarcinoma Reveals Antigen-Presenting Cancer-Associated Fibroblasts. *Cancer Discov*. 2019 Aug;9(8):1102-1123.

2) The title does suggest a broader scope “tumor microenvironment”. If the authors want to keep such title, additional study of at least the immune microenvironment is necessary.

Answer: Thanks for the suggestion. We changed the manuscript title to “The Lin28b/Wnt5a axis drives pancreas cancer through crosstalk between the cancer associated fibroblasts and tumor epithelium”.

-The study is mostly limited to mouse work. At some point there is mentioning of the squamous subtype of human pancreatic cancer. Adding more detail on how all these genes/proteins are expressed in human tumors (and their subtypes) would greatly advance the study.

Answer: We greatly appreciate the reviewer’s suggestion. To broaden our study, we isolated human PDAC cell lines from 2 patients and found that LIN28B was highly expressed in hPDAC1# but not in hPDAC2# (Fig.2m-n). Consistently with the data from mPDAC, LIN28B expression in human CAFs is induced by conditioned medium from hPDAC1# tumor cells in a Wnt5a-dependent manner (Fig.3l).

Previous study have showed that CD44⁺/LIN28B⁺ subpopulation cells which were isolated from human pancreatic cancer tissues expressed high levels of stemness marker genes and possessed higher migratory and invasive ability (Shao et al., 2022). To further study the role of Lin28b-Wnt5a axis in advanced human PDAC, we analyzed the expression profiles of the distinct molecular subtypes of human PDAC from proteogenomic analysis of 140 pancreatic cancers (Cancer Genome Atlas Research Network.,2017; Cao L et al., 2021) and found that tumors of the squamous subtype exhibited elevated expression of Lin28b (Supplementary Fig.5a and Supplementary Table 1). Moreover, expression of the Lin28b and Wnt5a were correlated with that of the squamous subtype signature (Supplementary Fig.5b-c and Supplementary Table 1), underscoring the association between Lin28b-Wnt5a axis and squamous subtype tumors.

Shao Y, Zhang L, Cui L, Lou W, Wang D, Lu W, Jin D, Liu T. LIN28B suppresses microRNA let-7b expression to promote CD44⁺/LIN28B⁺ human pancreatic cancer stem cell proliferation and invasion. *Am J Cancer Res.* 2015 Aug 15;5(9):2643-59.

Cancer Genome Atlas Research Network. Integrated genomic characterization of pancreatic ductal adenocarcinoma. *Cancer Cell.* 2017;32(2):185–203.e13.

Cao L, Huang C, Cui Zhou D, Hu Y, Lih TM, Savage SR, Krug K, Clark DJ, Schnaubelt M, Chen L, da Veiga Leprevost F, Eguez RV, Yang W, Pan J, Wen B, Dou Y, Jiang W, Liao Y, Shi Z, Terekhanova NV, Cao S, Lu RJ, Li Y, Liu R, Zhu H, Ronning P, Wu Y, Wyczalkowski MA, Easwaran H, Danilova L, Mer

AS, Yoo S, Wang JM, Liu W, Haibe-Kains B, Thiagarajan M, Jewell SD, Hostetter G, Newton CJ, Li QK, Roehrl MH, Fenyö D, Wang P, Nesvizhskii AI, Mani DR, Omenn GS, Boja ES, Mesri M, Robles AI, Rodriguez H, Bathe OF, Chan DW, Hruban RH, Ding L, Zhang B, Zhang H; Clinical Proteomic Tumor Analysis Consortium. Proteogenomic characterization of pancreatic ductal adenocarcinoma. *Cell*. 2021 Sep 16;184(19):5031-5052.e26.

-Several analyses need to be refined.

1) Markers need to be assessed by double labelling to ascertain the cell type that is studied (e.g. for pKi67 – one cannot just assume that it are tumour cells that proliferate)

Answer: We stained CK19 as PDAC marker in Fig.5f, Fig.5n, Fig.5s and Fig.6r.

2) How are tumours exactly weighed after orthotopic grafting of the tumour cells?

Answer: Tumor burden was measured by establishing the gross wet weight of the pancreas. We also changed “tumor weight” to “pancreas weight” in the legend of figure.

3) KO of genes need to be validated.

Answer: Knocking-out efficiency was shown in Fig.4c-d, Fig.6m, Supplementary Fig.1e-h, Supplementary Fig.2g-h, Supplementary Fig.2k-m, Supplementary Fig.3a, Supplementary Fig.3d and Supplementary Fig.9b.

4) What is the receptor for Pcsk9 and is it expressed in the tumour cells? This is essential to show.

Answer: PCSK9 regulates plasma LDL-cholesterol level by binding to LDL receptor (LDLR) and targeting it for lysosomal degradation in cells (Lagace TA, 2014). In consequence, the increased plasma LDL-c levels could cause hypercholesterolemia that could mediate tumor cell growth (Mahboobnia et al. 2021). To test the level of Ldlr, we treated mPDAC cells with recombinant Pcsk9 protein and found that the protein level of Ldlr was decreased after the treatment (Supplementary Fig.9a).

Lagace TA. PCSK9 and LDLR degradation: regulatory mechanisms in circulation and in cells. *Curr Opin Lipidol*. 2014 Oct;25(5):387-93.

Mahboobnia K, Pirro M, Marini E, Grignani F, Bezsonov EE, Jamialahmadi T, Sahebkar A. PCSK9 and cancer: Rethinking the link. *Biomed Pharmacother*. 2021 Aug;140:111758.

5) What would have caused the divergence of 3 tumour cell lines from one genetically identical mouse line? Investigating this will add an extra level of information on tumor heterogeneity.

Answer: We appreciate the reviewer's concern on this matter and understand the importance of the question. The three mice, albeit of having the same two genetic mutations in Kras and Trp53 genes, have different genetic aberrations with different epigenetic, post-transcriptional and post-translational heterogeneities. Also, recent review paper (Hingorani SR, Nat Rev Cancer. 2023) mentioned that pancreatic ductal adenocarcinomas are distinguished by their robust desmoplasia, or fibroinflammatory response. Dominated by non-malignant cells, the mutated epithelium must therefore combat, cooperate with or co-opt the surrounding cells and signaling processes in its microenvironment. The expression difference is expected and is useful to delineate common gene expression signatures associated with the Kras and Trp53 mutations in mice. To address this question, we also performed the whole exome sequencing in these two different cell lines (different subtype based on our RNA-seq profiling) and we found that even they shared the same hot spot genetic mutations, and most SNPs are the same. These two cell lines still have some distinct unique SNPs as following (see below). Also, we found that they have some unique insertions and deletions which are the adaptive evolution as we expected. The detailed results from the whole exome sequencing experiments are deposited in the NCBI's Gene Expression Omnibus (GSE217378).

- Beta-catenin is mentioned as a Wnt receptor which is not correct.

Answer: We have corrected “ β -catenin is the receptor for Wnt-5a” into “ β -catenin is an intracellular signal transducer in the Wnt pathway” in our manuscript.

- The discussion mentions elements that were not obvious in the results section, such as the presumed role of cholesterol.

Answer: The effect of cholesterol on tumor growth was shown in Supplementary Fig.9c.

Reviewer #3 (Remarks to the Author):

In this research manuscript, the authors present evidence that there is a feedback loop where LIN28B expression in pancreatic ductal adenocarcinoma (PDAC) induces WNT5A expression that in turn induces LIN28B expression in cancer associated fibroblasts (CAFs) that in turn secretes the cytokine PCSK9 that acts on the cancer cells.

Overall, this research is quite solid with appropriate controls and well-justified conclusions that uses multiple methods to reach their conclusions.

Overall summary:

Initially, they perform immunohistochemistry on a 90 patient cohort that revealed that LIN28B expression in cancer cells and stroma are tightly correlated where LIN28B high expression is correlated with poor prognosis. This association is seen in multiple cancer types where LIN28A/LIN28B expression is almost invariably correlated with poor survival. However, the connection to the stroma and ultimately CAFs is novel.

Next, they establish that a secreted factor from LIN28B-expressing cancer cells induces LIN28B expression in CAFs using biochemical and cell line approaches where the secretome analysis using CRISPR modified cell lines that the soluble factor was WNT5A where recombinant WNT5A can induce LIN28B expression in CAFs.

Further work showed that LIN28B expression in CAFs upregulated a number of genes. This result is consistent with the widely published result that LIN28B negatively regulates biogenesis of the tumor suppressor let-7 microRNA family. The cytokine PCSK9 was selected for further study as this gene is predicted to be a let-7 regulated gene. The authors show both overexpression and inhibition experiments showing that PCSK9 is regulated by let-7. Further work shows that LIN28/let-7 likely regulates PCSK9 at the level of translational inhibition rather than mRNA degradation.

Minor points:

1. Figure 5E, the y-axis is miRNA I assume. Also, it is not clear what this data is normalized to.

Answer: We greatly appreciate the reviewer's suggestion. We correct the y-axis label in Figure 6E.

The miRNA was normalized to snRNA U6 and the information was also added in the label of Figure 6E.

2. For Figure 5 and the text, the authors most likely need to rename things. Initially, I thought LIN28-WT and LIN28B-MU lines that had both endogenous LIN28B expression and overexpression of the indicated protein. It turns out that this was a LIN28B CRISPR/Cas9 knockout line where an add back experiment was done. This is confusing.

Answer: Yes. We corrected the manuscript into “The CRISPR/Cas9-resistant wild-type Lin28b (flag-Lin28b-WT(r)) and mutant Lin28b (flag-Lin28b-MU(r)) were overexpressed in Lin28b-null cells”.

3. Please update the manuscript using the approved gene nomenclature (e.g., human LIN28B, WNT5A,... and mouse Lin28b, Wnt5a, etc...) <https://www.genenames.org/> and <http://informatics.jax.org/mgihome/nomen/>

Answer: Thanks for the constructive suggestions. We corrected the genes name according to the reviewer’s suggestion.

REVIEWER COMMENTS

Reviewer #1 (Remarks to the Author):

In this revised version of the manuscript, the authors have addressed some but not all the concerns raised by this reviewer. Overall, I think that the revision has improved the quality of the manuscript.

However, important concerns remain especially when it comes to the mechanisms through which Lin28b promotes PC progression via either cell autonomous or non-cell autonomous effects. One important drawback of the study, which has not been adequately addressed by the authors, is the mechanisms of action of PCSK9. No relevant mechanistic insight has been provided.

More data have been generated to support the potential role of Wnt5a in driving Lin28b expression in CAFs but they are not convincing to me. A thorough assessment of the canonical wnt pathway activation downstream of wnt5a should be performed. That would include performing a reporter assay (tcf/lef) following exposure of CAFs to either conditioned media (CM) or rWnt5a. Also, data about promoter occupancy and beta-catenin stabilization are pretty weak (Figure 4i-j). Related to the same set of displays, what is the reason for not evaluating promoter occupancy after treatment with rWnt5a? Is the need for CM suggestive that other components are needed for the induction of the pathway? As also suggested during the first revision, I would get away from experiments using conditioned media as they are inherently "dirty" because many factors could contribute to the observed phenotypes. And even if the CM is prepared from cells deficient for the protein of interest, we have to acknowledge that the KO of that protein may alter the secretome in more complex ways. Something else that has not been explored, and should be, is the effect of Lin28b perturbation on CAFs phenotypes. Do the lin28b+ and lin28b- CAFs have different proliferation and engraftment capacity? The CAFs are immortalized with SV40, which means that they express a very immunogenic protein. Differences in take rate and proliferation would probably lead to different immunogenic effect in vivo, translated then in different growth rate of the tumor cells. I do see ki67 signal in the stromal compartment, which is not expected as CAFs rarely if never stain for ki67 in human or mouse autochthonous tissues. I would suggest performing the same experiments but in an immunodeficient setting.

Few other comments below

- Figure 6u. There is no control arm for this experiment (PCSK9 deficient cells) to evaluate/assess the specificity of drug towards the target.
- Dataset should be better annotated. It is very difficult to interpret the data.
- Dataset related to iCAFs/myCAFs score. I cannot find information on how the score has been defined.
- Still not clear how the conditioned media for proteomic analysis has been prepared. The legend to the figure and the materials and methods section describes two different procedures. That is quite important to clarify. If the conditioned media has been harvested after 24 hours in serum-free medium that raises concerns about the stressed state and viability of CAFs. They do not like to sit without fbs for 24 hours. If the exposure to the conditioned media from Lin28b+ cancer cells is so important for the induction of lin28b in CAFs (several days of conditioning), what about the maintenance of lin28b expression. How long does it take for the effect of the CM to vanish, i.e., are you sure that lin28b is still induced after 24 hours without inducing medium. Why not comparing lin28b+ vs - CAFs for

secretome analysis?

- There is no clear response to my concerns about the differences (as opposed to the current literature) in observing a prominent cell autonomous effect of lin28b

Reviewer #2 (Remarks to the Author):

Most of my comments have been addressed. I have no further comments besides:

-The authors responded to my remarks on the myCAFs and their heterogeneity. A shortened version of this response would be suitable to integrate in the discussion.

-A double labeling Ki67 and panCK in Fig5S would have been more accurate to perform quantifications (in Fig 5T). In other figures, such double immunofluorescence was done so it could be done here as well.

-throughout the manuscript the authors use 'posotive' instead of 'positive'

Reviewer #3 (Remarks to the Author):

This manuscript was a strong initial submission. The response to reviewers was thorough and strengthened the manuscript. It is very solid science in my opinion.

REVIEWER COMMENTS

Reviewer #1 (Remarks to the Author):

In this revised version of the manuscript, the authors have addressed some but not all the concerns raised by this reviewer. Overall, I think that the revision has improved the quality of the manuscript.

However, important concerns remain especially when it comes to the mechanisms through which Lin28b promotes PC progression via either cell autonomous or non-cell autonomous effects. One important drawback of the study, which has not been adequately addressed by the authors, is the mechanisms of action of PCSK9. No relevant mechanistic insight has been provided.

Answer: PCSK9 regulates plasma LDL-cholesterol level by binding to LDL receptor (LDLR) and targeting it for lysosomal degradation in cells (Lagace TA, 2014). In consequence, the increased plasma LDL-c levels could cause hypercholesterolemia that could mediate tumor cell growth (Mahboobnia et al. 2021). To test the level of Ldlr, we treated mPDAC cells with recombinant Pcsk9 (r-Pcsk9) protein and found that the protein level of Ldlr was decreased after the treatment (Supplementary Fig.10a). Then we overexpressed Ldlr-expression plasmid in mPDAC cells and found that overexpression of Ldlr diminished the effect of r-Pcsk9 protein on cell viability (Supplementary Fig.10c-d). In addition, we examined the efficacy of PCSK9 inhibitor alirocumab, a monoclonal antibody that binds circulating PCSK9 and blocks its interactions with surface LDLR, Alirocumab blocked r-Pcsk9's effect on tumor growth (Supplementary Fig.10e), suggesting that Pcsk9 promotes tumor dependent on its binding with Ldlr.

The novelty of this paper is discovering the tumor non-cell autonomous function of Lin28b in PDAC and the Lin-28B-Wnt-5a Axis in tumor epithelial regulates the secretion of PSCK9 from CAFs, which reversely supports tumor growth. To be honest, PSCK9 is the target of our next story, we want to deeply investigate the mechanism how PSCK9 be regulated, and we will translate our findings (PSCK9 inhibitor) to treat Lin28b positive PDAC in some specific PDAC subtype.

Lagace TA. PCSK9 and LDLR degradation: regulatory mechanisms in circulation and in cells. *Curr Opin Lipidol.* 2014 Oct;25(5):387-93.

Mahboobnia K, Pirro M, Marini E, Grignani F, Bezsonov EE, Jamialahmadi T, Sahebkar A. PCSK9 and cancer: Rethinking the link. *Biomed Pharmacother.* 2021 Aug;140:111758.

More data have been generated to support the potential role of Wnt5a in driving Lin28b expression in CAFs but they are not convincing to me. A thorough assessment of the canonical wnt pathway activation downstream of wnt5a should be performed. That would include performing a reporter assay (tcf/lef) following exposure of CAFs to either conditioned media (CM) or rWnt5a. Also, data about promoter occupancy and beta-catenin stabilization are pretty weak (Figure 4i-j). Related to the same set of displays, what is the reason for not evaluating promoter occupancy after treatment with rWnt5a? Is the need for CM suggestive that other components are needed for the induction of the pathway? As also suggested during the first revision, I would get away from experiments using conditioned media as they are inherently "dirty" because many factors could contribute to the observed phenotypes. And even if the CM is prepared from cells deficient for the protein of interest, we have to acknowledge that the KO of that protein may alter the secretome in more complex ways.

Answer: Firstly, the promoter occupancy and beta-catenin stabilization experiment are Figure 4k-l and 4i, and we tested the promoter occupancy after treatment with rWnt5a in fig4k-l (green column vs. blue column). We also performed TCF/LEF reporter assay and found both Lin28b^{high} tumors-CM and r-Wnt5a can activated TCF reporter gene in CAFs (Fig.4m).

Secondly, we provided the quantification and statistics for the western blots showing beta-catenin stabilization in Fig.4i.

Most importantly, reviewer1 has big concern about condition medium, however, there are tons of studies shown that condition media can be used in co culture system, which almost been the standard culture system in pancreatic cancer study. For example:

- 1) Squamous trans-differentiation of pancreatic cancer cells promotes stromal inflammation (Elife) PMID: 32329713
- 2) Distinct populations of inflammatory fibroblasts and myofibroblasts in pancreatic cancer (JEM) PMID: 28232471
- 3) Altered glucose metabolism and proteolysis in pancreatic cancer cell conditioned myoblasts: searching for a gene expression pattern with a microarray analysis of 5000 skeletal muscle genes (GUT) PMID: 15247186
- 4) Hyaluronic acid fuels pancreatic cancer cell growth (Elife) PMID: 34951587

Something else that has not been explored, and should be, is the effect of Lin28b perturbation on CAFs phenotypes. Do the lin28b+ and lin28b- CAFs have different proliferation and engraftment capacity? The CAFs are immortalized with SV40, which means that they express a very immunogenic protein. Differences in take rate and proliferation would probably lead to different immunogenic effect in vivo, translated then in different growth rate of the tumor cells. I do see ki67 signal in the stromal compartment, which is not expected as CAFs rarely if never stain for

ki67 in human or mouse autochthonous tissues. I would suggest performing the same experiments but in an immunodeficient setting.

Answer: As reviewer said Ki67 staining is rarely shown in stroma of human or GEMM model, but in our system, he/she saw some staining of Ki67 in stroma. This is an orthotopic mouse model, tumor growth is much faster than GEMM or even human primary tumor, and we should say it has some limitation but not a lot. The CAFs we generated and stained in our paper is from 15376 Tumor, which is a basal like subtype ikras cell line. As reviewer known, most PDAC from human patients or GEMM model are classical subtype (we harvested ten different cell lines from ikras model, only one cell line shown basal like). We hypothesize but not sure, the CAFs from different subtype of PDAC have different proliferation rate. In addition, the proliferation rate of CAFs was not significantly changed by Lin28b (Supplementary Fig.3d-e).

As to SV40, most groups and good studies are using immortalized CAFs to investigate the crosstalk between CAFs and tumor epithelium. For example:

- 1) High throughput screen in a co-culture model to uncover therapeutic strategies to potentiate the cancer-inhibiting properties of the tumor-stroma in pancreatic cancer (BioRxiv) doi.org/10.1101/2022.03.11.483991
- 2) Pancreatic stellate cells support tumour metabolism through autophagic alanine secretion (Nature) PMID: 27509858
- 3) Mesothelial cell-derived antigen-presenting cancer-associated fibroblasts induce expansion of regulatory T cells in pancreatic cancer (Cancer cell) PMID: 35523176

Also, we are not studying the immunology related question in our paper, and proved our idea *in vitro* (Fig.5a-d). I think it is obvious that immune cell lineages are not involved. But we still thanks reviewer1's idea, we can go deep and see how immune cells are involved in different subtype and crosstalk with CAFs in our future work.

Few other comments below

- Figure 6u. There is no control arm for this experiment (PCSK9 deficient cells) to evaluate/assess the specificity of drug towards the target.

Answer: Alirocumab is a specific drug for PCSK9 since it's a monoclonal antibody that binds circulating PCSK9 and blocks its interactions with surface LDLR.

- Dataset should be better annotated. It is very difficult to interpret the data.

Answer: Yes, we rewrote the description of dataset to make it clearer.

- Dataset related to iCAFs/myCAFs score. I cannot find information on how the score has been defined.

Answer: Yes, we added the score definition method in statistical analysis part.

- Still not clear how the conditioned media for proteomic analysis has been prepared. The legend to the figure and the materials and methods section describes two different procedures. That is quite important to clarify. If the conditioned media has been harvested after 24 hours in serum-free medium that raises concerns about the stressed state and viability of CAFs. They do not like to sit without fbs for 24 hourse. If the exposure to the conditioned media from Lin28b+ cancer cells is so important for the induction of lin28b in CAFs (several days of conditioning), what about the maintenance of lin28b expression. How long does it take for the effect of the CM to vanish, i.e., are you sure that lin28b is still induced after 24 hours without inducing medium. Why not comparing lin28b+ vs - CAFs for secretome analysis?

Answer: Firstly, the maintenance of lin28b expression was tested in our first-round revision. As shown in Supplementary Fig.4a, Lin28b is induced after 24 hours without inducing medium. Then the viability of CAFs was not significantly changed by serum-free medium treatment (Supplementary Fig.4b). We rewrote the figure legends to make it clearer.

- There is no clear response to my concerns about the differences (as opposed to the current literature) in observing a prominent cell autonomous effect of lin28b

Answer: We have used several human cell lines and found PANC-1 and PANC03.27 which expressed high level of LIN28B do dependency on LIN28B's function as reported before (Supplementary Fig.7a-b). Therefore we cannot say Lin28b has no cell autonomous function. But we also observed the non-cell autonomous in the orthotopic mouse model which was generated in *FSP-Cre;Lin28b^{fl/fl}* mice. The discrepancy between the findings in our study and previous reports may result from different tumor model we used. And we have added it in the discussion part of our manuscript.

Reviewer #2 (Remarks to the Author):

Most of my comments have been addressed. I have no further comments besides:

-The authors responded to my remarks on the myCAFs and their heterogeneity. A shortened version of this response would be suitable to integrate in the discussion.

Answer: Yes, we have added it in the discussion part of our manuscript.

-A double labeling Ki67 and panCK in Fig5S would have been more accurate to perform quantifications (in Fig 5T). In other figures, such double immunofluorescence was done so it could be done here as well.

Answer: Yes, we performed double immunofluorescence of CK19 and Ki67 in Fig.5q.

-throughout the manuscript the authors use 'posotive' instead of 'positive'

Answer: We have corrected it in our manuscript.

Reviewer #3 (Remarks to the Author):

This manuscript was a strong initial submission. The response to reviewers was thorough and strengthened the manuscript. It is very solid science in my opinion.

Answer: We appreciate the reviewer very much for his/her positive comments.

REVIEWERS' COMMENTS

Reviewer #1 (Remarks to the Author):

The authors have done a commendable job in addressing reviewers' comments. I have no further questions/comments

Reviewer #2 (Remarks to the Author):

My comments have been properly addressed.

Reviewer #1 (Remarks to the Author):

The authors have done a commendable job in addressing reviewers' comments. I have no further questions/comments

Reviewer #2 (Remarks to the Author):

My comments have been properly addressed.

Response: On behalf of my co-authors, we would like to express our great appreciation to reviewers for their positive comments on our manuscript.